# Optimality and Stability in Federated Learning: A Game-theoretic Approach

**Kate Donahue**
Department of Computer Science
Cornell University
kdonahue@cs.cornell.edu

**Jon Kleinberg**
Departments of Computer Science
and Information Science
Cornell University
kleinber@cs.cornell.edu

## Abstract

Federated learning is a distributed learning paradigm where multiple agents, each only with access to local data, jointly learn a global model. There has recently been an explosion of research aiming not only to improve the accuracy rates of federated learning, but also provide certain guarantees around social good properties such as total error. One branch of this research has taken a game-theoretic approach, and in particular, prior work has viewed federated learning as a hedonic game, where error-minimizing players arrange themselves into federating coalitions. This past work proves the existence of stable coalition partitions, but leaves open a wide range of questions, including how far from optimal these stable solutions are. In this work, we motivate and define a notion of optimality given by the average error rates among federating agents (players). First, we provide and prove the correctness of an efficient algorithm to calculate an optimal (error minimizing) arrangement of players. Next, we analyze the relationship between the stability and optimality of an arrangement. First, we show that for some regions of parameter space, all stable arrangements are optimal (Price of Anarchy equal to 1). However, we show this is not true for all settings: there exist examples of stable arrangements with higher cost than optimal (Price of Anarchy greater than 1). Finally, we give the first constant-factor bound on the performance gap between stability and optimality, proving that the total error of the worst stable solution can be no higher than 9 times the total error of an optimal solution (Price of Anarchy bound of 9).

## 1 Introduction

Recent advances of machine learning techniques has made it possible to apply powerful prediction algorithms to a variety of domains. However, in real-world situations, data is often distributed across multiple locations and cannot be combined to a central repository for training. For example, consider patient medical data located at hospitals or student educational data at different schools. In each case, the individual agents (hospitals or schools) who hold the data wish to find a model that minimizes their error. However, the data at each location may be insufficient to train a robust model. Instead, the agents may prefer to build a model using data from multiple agents: multiple hospitals or schools. Collectively, the combined data may be able to produce a model with much higher accuracy, providing more powerful predictions to each agent and increasing overall welfare. However, it may be infeasible to transfer the data to some coordinating entity to build a global model: privacy, data size, and data format are all possible reasons that would make transferring data not a reasonable solution.

Federated learning is a novel distributed learning paradigm that aims to solve this problem (McMahan et al. [2016]). Data remains at separate local sites, which individual agents use to learn local model parameters or parameter updates. Then, only the parameters are transferred to the coordinating entity

(for example, a technology company), which averages together all of the parameters in order to form a single global model, which all of the agents use. Federated learning is a rapidly growing area of research (Li et al. [2020], Kairouz et al. [2019], Lim et al. [2020]).

However, research has also noted that federated learning, in its traditional form, may not be the best option for each agent (Yu et al. [2020], Bagdasaryan and Shmatikov [2019], Li et al. [2019], Mohri et al. [2019]). In the real world, agents may differ in their true distribution: the true model of patient outcomes at hospital $A$ may differ from the true model at hospital $B$, for example. If these differences are large enough, federating agents may see their error increase under certain situations, potentially even beyond what they would have obtained with only local learning. For example, a player with relatively few samples may end up seeing its model "torqued" by the presence of a player with many samples. For this reason, agents may not wish to federate with every other potential agent.

Instead, each agent faces a choice: given the costs and benefits of federating with different players, it must determine which of the exponentially many combinations of players it would prefer to federate with. Simultaneously, every other agent is also attempting to identify and join a federating group that it prefers - and agents may have conflicting preferences. Prior work (Donahue and Kleinberg [2021], Hasan [2021]) has formulated this problem as a *hedonic game*, which each player derives some cost (error) from the coalition they join. The aim of such research has been to identify partitions of players that are *stable* against deviations, for varying definitions of stability. A hedonic game in general may not have any stable arrangements, so the area's contributions in the analysis of stability adds valuable insight into the incentives of federating agents.

However, this framework also leaves open multiple game theoretic questions. While the federating agents have individual incentives to reduce their error, society as a whole also has an interest in minimizing the overall error. In the school example, individual schools wish to find coalitions that work well on their own sub-populations, while the overall district or state may have an interest in finding an overall set of coalitions that minimizes the overall error. This analysis of a coalition partition's overall cost falls under the game theoretic notion of *optimality*.

One natural question relates to the tension between these two goals: the self-interested goal of the individual actors (stability) and the overall goal of reducing total cost (optimality). Given a that set of self-interested agents has found a stable solution, how far from optimal could it be? This is reflected by the *Price of Anarchy* of a game, the canonical approach to study optimality and stability jointly (Papadimitriou [2001], Koutsoupias and Papadimitriou [1999]). The Price of Anarchy (PoA) is a ratio where the numerator is equal to the highest-cost stable arrangement and the denominator is equal to the lowest-cost arrangement (the optimal arrangement). It is lower bounded by 1, a bound that it achieves only if all stable arrangements are optimal. A higher Price of Anarchy value implies a greater trade off between stability and optimality, and bounding the Price of Anarchy for a particular game puts a limit on this trade-off. Federated learning is a situation where questions of stability have been analyzed, but to our knowledge there has been no systematic analysis of the Price of Anarchy in a model of federated learning.

**The present work: A framework for optimality and stability in federated learning** In this work, we make two main contributions to address this gap. First, we provide an efficient, constructive algorithm for calculating an optimal federating arrangement. Secondly, we prove the first-ever constant bound on the Price of Anarchy for this game, showing that the worst stable arrangement is no more than 9 times the cost of the best arrangement.

We begin Section 4 by defining optimality, drawing on a notion of weighted error derived from the standard objective in federated learning literature. The main contribution of this section is an efficient, constructive algorithm for calculating an optimal arrangement, along with a proof of its optimality. However, as demonstrated in Section 5, optimality and stability are not always simultaneously achieved. This section analyzes the Price of Anarchy, which measures how far from optimal the worst stable arrangement can be. First, we demonstrate that the optimal arrangement is not always stable. Next, we show that there exist sub-regions where the Price of Anarchy is equal to 1. Finally, this section proves an overall Price of Anarchy bound of 9, the first constant bound for this game.

It is worth emphasizing that, beyond the Price of Anarchy bound itself, part of the contribution of this work is the optimization and analysis to produce this bound. The proofs for this contribution are modular and illuminate multiple properties about the broader federated learning game under study.

As such, these contributions could be useful for further investigating this model. For example, the modular structure of our proof is what enables us to establish stronger bounds for certain sub-cases.

The results in this paper are theoretical and do not depend on any experiments or code. However, while writing the paper, we found it useful to write and work with code to check conjectures. Some of that code is publicly available at `github.com/kpdonahue/model_sharing_games`.

## 2 Related work

**Federated learning**   As we mentioned previously, federated learning has recently seen numerous advances. In this section, we highlight a few papers in federated learning that are especially related to our work.

The idea that agents might differ in their true models (that data might be generated non-i.i.d. across multiple agents) is commonly acknowledged in the federated learning literature. For example, Yu et al. [2020], Bagdasaryan and Shmatikov [2019] empirically demonstrate that federated learning, and especially privacy-related additions, can cause a wide disparity in error rates. Some techniques have been developed specifically towards this problem. For example, hierarchical federated learning adds an additional layer of hierarchical structure to federated learning, which could be used to reduce latency or to cluster together similar players (Lin et al. [2018], Liu et al. [2020]). Many other works also relate to clustering, such as (Lee et al. [2020], Sattler et al. [2020], Shlezinger et al. [2020], Wang et al. [2020], Duan et al., Jamali-Rad et al. [2021], Caldarola et al. [2021]). Nagalapatti and Narayanam [2021] uses tools from cooperative game theory to selectively combine gradient updates from multiple agents. These works, which tend to be more applied than our work, may also differ in that they analyze situations where additional information is known, such as the data distribution at each location.

Other work aims to improve accuracy rates by selecting acquiring additional data (Blum et al. [2017], Duan et al. [2021]). Some papers specifically analyze federated learning for high-stakes situations such as medical settings (Xia et al. [2021], Guo et al. [2021], Vaid et al. [2021], Kumar et al. [2021], Zhang et al. [2021]). In general, all of these works have the goal of reducing the average error over all federating agents, which we will use to motivate our definition of optimality in later sections.

**Game theory in federated learning**   The closest work to this current paper is our prior paper Donahue and Kleinberg [2021], which we discuss in greater detail in Section 3. Another paper using hedonic game theory to analyze federated learning games is Hasan [2021], which gives conditions for Nash stability in federated learning. Other works analyze the incentives of players to contribute resources towards federated learning: Blum et al. [2021] analyzes fairness and efficiency in sampling additional points for federated learning and Le et al. [2021] analyzes incentives for agents to contribute computational resources in federated learning. Interestingly, multiple works take a game theoretic approach towards coalition formation in cloud computing, but with the aim of minimizing some cost besides error, such as electricity usage Guazzone et al. [2014], Anglano et al. [2018].

## 3 Model and assumptions

We assume that there are $M$ total agents (sometimes referred to as players). Each agent $i \in [M]$ has drawn $n_i$ data points from their true local distribution $g(\theta_i)$, where $\theta_i$ are their true local parameters and $g(\cdot)$ is some true labeling function. A player's goal is to learn a model with low expected error on its own distribution. If a player opts for local learning, then it uses its local estimate of these parameters $g(\hat{\theta}_i)$ to predict future data points, obtaining error $err_i(\{i\})$. If a set of players $C$ are federating together, we say that they are in a *coalition* or *cluster* together. They combine their local estimates of parameters into a single federated estimate, governed by the weighted average of their parameters:

$$\hat{\theta}_C = \frac{1}{\sum_{i \in C} n_i} \cdot \sum_{i \in C} n_i \cdot \hat{\theta}_i \tag{1}$$

A federating player $i$ obtains error $err_i(C)$: note that this value may differ between players in the same coalition. For example, if player $j$ has more samples than player $k$, then $\hat{\theta}_C$ will be weighted more towards player $j$, meaning that player $j$ will have lower expected error than $k$.

The weighted average method in Equation 1 is commonly used in federated learning (McMahan et al. [2016]). Because it is the most straightforward method, it is sometimes called "vanilla" federated learning. Alternative ways of federation might involve customizing the model for individuals, as in domain adaptation. For example, Donahue and Kleinberg [2021] models three methods of federation: vanilla (called "uniform"), as well as two models of domain adaptation.

There are multiple reasons why we opted to analyze the federation method in Equation 1 in this work. First of all, this federation method is the most straightforward method, and as such it is the natural candidate to begin analysis. Secondly, this federation method is the most interesting to analyze technically. Domain adaptation serves to increase the incentives of an individual agent to participate in federation: it reduces the tension between an individual's incentives and the optimal overall arrangement. Because of this, for Price of Anarchy it is more valuable and challenging to explore the case in Equation 1, where incentives are more opposed.

## 3.1 Theoretical model of federation from Donahue and Kleinberg [2021]

Federated learning has been the subject of both applied and theoretical analysis; our focus here is on the theoretical side. In addition, for game theoretical reasoning to be feasible, we need a model that gives exact errors (costs) for each player, rather than bounds: these are needed in order to be able to argue that a certain arrangement is optimal, for example.

We opt to use the model developed in our prior work Donahue and Kleinberg [2021], which produces the closed-form error value seen in Lemma 1 below. While we work within this model, we emphasize that Donahue and Kleinberg [2021] asked different questions from this paper's focus: our prior work focused on developing the federated learning model and analyzing the stability of federating coalitions, while our current work analyzes optimality and Price of Anarchy.

**Lemma 1** (Lemma 4.2, from Donahue and Kleinberg [2021]). *Consider a mean estimation task as follows: player $j$ is trying to learn its true mean $\theta_j$. It has access to $n_j$ samples drawn i.i.d. $Y \sim \mathcal{D}_j(\theta_j, \epsilon_j^2)$, a distribution with mean $\theta_j$ and variance $\epsilon_j^2$. Given a population of players, each has drawn parameters $(\theta_j, \epsilon_j^2) \sim \Theta$ from some common distribution $\Theta$. A coalition $C$ federating together produces a single model based on the weighted average of local means (Eq. 1). Then, the expected mean squared error player $j$ experiences in coalition $C$ is:*

$$err_j(C) = \frac{\mu_e}{\sum_{i \in C} n_i} + \sigma^2 \cdot \frac{\sum_{i \in C, i \neq j} n_i^2 + \left(\sum_{i \in C, i \neq j} n_i\right)^2}{\left(\sum_{i \in C} n_i\right)^2} \tag{2}$$

*where $\mu_e = \mathbb{E}_{(\theta_i, \epsilon_i^2) \sim \Theta}[\epsilon_i^2]$ (the average noise in data sampling) and $\sigma^2 = Var(\theta_i)$ (the average distance between the true means of players).*

Note that Donahue and Kleinberg [2021] also analyzes a linear regression game with a similar cost function, though in this work we will restrict our attention to the mean estimation game.

We use some of the same notion and modeling assumptions as Donahue and Kleinberg [2021]. For example, we use $C$ to refer to a coalition of federating agents and $\Pi$ to refer to a collection of coalitions that partitions the $M$ agents. We will use $N_C$ to refer to the total number of samples present in coalition $C$: $N_C = \sum_{i \in C} n_i$. In a few lemmas we will re-use minor results proven in Donahue and Kleinberg [2021], citing them for completeness.

For technical assumptions, we assume number of samples $\{n_i\}$ is fixed and known by all. We also assume that the parameters $\mu_e, \sigma^2$ are approximately known: in particular, results will depend on whether the number of samples is larger or smaller than the critical threshold $\frac{\mu_e}{\sigma^2}$. We assume that a player does not know anything else about its own true parameters $\theta_i$ or the parameters of other players: for example, it does not know the true generating distribution $\Theta$ or if its true parameters are likely to lie far from the parameters of other players. We assume that each player has a goal of obtaining a model with low expected test error on its personal distribution, and that the federating coordinator is motivated to minimize some notion of total cost, but is otherwise impartial.

Finally, it is worth emphasizing key differences between this current work and Donahue and Kleinberg [2021]. The focus of Donahue and Kleinberg [2021] is defining a theoretical model of federated learning and analyzing the stability of such an arrangement. As such, it focuses solely on individual incentives, rather than overall societal welfare. On this other hand, this current work focuses on

discussions of optimality (overall welfare) and Price of Anarchy. Finally, this paper work is in some ways more general: while some results in Donahue and Kleinberg [2021] only allow players to have two different numbers of samples ("small" or "large"), every result in our work holds for arbitrarily many players with arbitrarily many different numbers of samples. This distinction is a function of the questions analyzed in each paper: questions of stability (as in Donahue and Kleinberg [2021]) are much harder to analyze for players with arbitrarily many different sizes.

## 4  Optimality

We will begin with the question of optimality. As motivation, it is useful to consider the objective function of most federated learning papers McMahan et al. [2016]:

$$\min_{\theta} err_w(\theta) = \sum_{i=1}^{M} p_i \cdot err_i(\theta) =^* \frac{1}{\sum_{i=1}^{M} n_i} \sum_{i=1}^{M} n_i \cdot err_i(\theta)$$

While the weights can be any $p_i > 0$, $\sum_{i=1}^{M} p_i = 1$, the $*$ equality reflects the common setting where they are taken to be the empirical average. In this work, we will take the empirical average as our cost function:

**Definition 1.** *A coalition partition $\Pi$ is optimal if it minimizes the weighted sum of errors across players, as defined below:*

$$f_w(\Pi) = \sum_{C \in \Pi} f_w(C) = \sum_{C \in \Pi} \sum_{i \in C} n_i \cdot err_i(C)$$

*We will say that a coalition partition $\Pi$ is in $OPT$ if it achieves minimal cost. Note that multiple partitions may achieve minimal cost, so $OPT$ is a set of partitions.*

Because $\Pi$ is a disjoint partition over the $M$ players, $f_w(\Pi)$ is simply the error $err_w(\theta)$ scaled by a constant. Therefore, minimizing $f_w(\Pi)$ is equivalent to minimizing the weighted average of errors.

Some machine learning papers modify the empirical average objective to achieve other goals. For example, Li et al. [2019], Mohri et al. [2019] consider variants where this goal is re-weighted in order to achieve certain fairness goals. Appendix A discusses other possible cost functions.

All of the above analysis holds for any model of federated learning. Lemma 2, below, gives the specific form of cost for federated learning using the model from Donahue and Kleinberg [2021]. The remaining analysis in this paper will assume this cost function. Proofs for results in this section are given in Appendix B.

**Lemma 2.** *Consider a partition $\Pi$ made up of coalitions $\{C_i\}$. Then, using the error form given in Equation 2, the total cost of $\Pi$ is given by*

$$f_w(\Pi) = \sum_{C \in \Pi} \left\{ \mu_e + \sigma^2 \cdot N_C - \sigma^2 \frac{\sum_{i \in C} n_i^2}{N_C} \right\}$$

The two most common arrangements in machine learning tasks are local learning (which we will denote by $\pi_l$) and the federation in the *grand coalition* ($\pi_g$), where all of the players are federating together in a single coalition. However, Lemmas 3 and 4 demonstrate that either of these could perform arbitrarily poorly as compared the cost-minimizing (optimal) arrangement.

**Lemma 3.** $\forall \rho > 1$, *there exists a setting where local learning results in average error more than $\rho$ times higher than optimal:* $\frac{f_w(\pi_l)}{f_w(OPT)} > \rho$.

**Lemma 4.** $\forall \rho > 1$, *there exists a setting where federating in the grand coalition results in average error more than $\rho$ times higher than optimal:* $\frac{f_w(\pi_g)}{f_w(OPT)} > \rho$.

A priori, finding a partition of players that minimizes total cost seems extremely challenging. There are exponentially many options for partitions, and two lemmas above have shown that either of the most common choices could be arbitrarily far from optimal. However, the next section will provide an efficient, constructive algorithm to calculate an optimal partition of players into federating coalitions.

## 4.1 Calculating an optimal arrangement

The main contribution of this section is Theorem 1, which gives an algorithm for minimizing the total weighted error of the federating agents.

**Theorem 1.** *Consider a set of players $\{n_i\}$. An optimal partition $\Pi$ can be created as follows: first, start with every player doing local learning. Then, begin by grouping the players together in ascending order of size, stopping when the first player would increase its error by joining the coalition from local learning. Then, the resulting partition $\Pi$ is optimal.*

Though the algorithm in Theorem 1 is straightforward, proving the optimality of the resulting partition $\Pi$ requires several sub-lemmas. Each sub-lemma is a building-block that describes certain operations that either increase or decrease total cost. The proof of Theorem 1 largely consists of sequentially using these sub-lemmas in order to demonstrate the optimality of the calculated partition.

**Statement and description of supporting lemmas** First, Lemma 5 demonstrates a close relationship between movements of players that reduce total cost and movements of players that are in that player's self-interest (recall that players always wish to minimize their expected error). Specifically, it shows that a player wishes to join a coalition from local learning if and only if that move would reduce total cost for the entire partition.

**Lemma 5** (Equivalence of player preference and reducing cost). *Take any coalition $Q$ and any player $j$. Then, a player wishes to join that coalition (from local learning) if and only if doing so would reduce total cost. That is,*

$$f_w(\{n_j\}) + f_w(Q) \geq f_w(\{n_j\} \cup Q) \quad \Leftrightarrow \quad err_j(\{n_j\}) \geq err_j(\{n_j\} \cup Q)$$

Next, Lemma 6 shows that "swapping" the roles of two players (one doing local learning, one federating in a coalition) reduces total cost when the larger player is removed to local learning.

**Lemma 6** (Swapping). *Take any set $Q$ including a player $n_j > n_k$, where the player $n_k$ is doing local learning. Then, swapping the roles of players $k$ and $j$ always decreases total cost.*

$$f_w(Q \cup \{n_j\}) + f_w(\{n_k\}) > f_w(Q \cup \{n_k\}) + f_w(\{n_j\})$$

Lemmas 7 and 8 give results for when players are incentivized to leave or join a particular coalition: they show that such incentives are monotonic in the size of the player. By Lemma 5, these results also show the monotonicity of cost-reducing operations. Note that these lemmas are not equivalent: they differ in whether the reference player $j$ is already in the coalition or not.

**Lemma 7** (Monotonicity of joining). *If a player of size $n_j$ would prefer local learning to joining a coalition $Q$, then any player of size $n_k \geq n_j$ also prefers local learning to joining the same coalition. That is, for $n_k \geq n_j$,*

$$err_j(Q \cup \{n_j\}) \geq err_j(\{n_j\}) \quad \Rightarrow \quad err_k(Q \cup \{n_k\}) \geq err_k(\{n_k\})$$

*Conversely, if a player $j$ wishes to join $Q$, then any other player of size $n_k \leq n_j$ would have also wanted to join. That is, for $n_j \geq n_k$,*

$$err_j(Q \cup \{n_j\}) \leq err_j(\{n_j\}) \quad \Rightarrow \quad err_k(Q \cup \{n_k\}) \leq err_k(\{n_k\})$$

**Lemma 8** (Monotonicity of leaving). *Take any coalition $Q$. Then, if any player $j \in Q$ of size $n_j$ wishes to leave $Q$ for local learning, then any player of size $n_k \geq n_j$ also wishes to leave for local learning. That is, for $n_k \geq n_j$*

$$err_j(Q) \geq err_j(\{n_j\}) \quad \Rightarrow \quad err_k(Q) \geq err_k(\{n_k\})$$

*Conversely, if a player $j \in Q$ of size $n_j$ does* not *wish to leave $Q$ for local learning, then any player $k \in Q$ of size $n_k \leq n_j$ also does not wish to leave. That is, for $n_k \leq n_j$*

$$err_j(Q) \leq err_j(\{n_j\}) \quad \Rightarrow \quad err_k(Q) \leq err_k(\{n_k\})$$

All of the above lemmas have analyzed situations where a single player is moving between coalitions. Lemma 9 analyzes cases where multiple players are rearranged simultaneously. Specifically, it provides an algorithm for combining together two separate groups (and then removing certain players) that is guaranteed to keep constant or reduce total cost.

| Coalition structure | $err_a(\cdot), n_a = 1$ | $err_b(\cdot), n_b = 8$ | $err_c(\cdot), n_c = 15$ | $f_w(\Pi)$ | $err_w(\Pi)$ |
|---|---|---|---|---|---|
| $\{a,\},\{b\},\{c\}$ | 10 | 1.25 | 0.667 | 30 | 1.25 |
| $\{a\},\{b,c\}$ | 10 | 1.285 | 0.677 | 30.435 | 1.268 |
| $\{a,c\},\{b\}$ | 2.382 | 1.25 | 0.633 | 21.875 | 0.911 |
| $\{a,b\},\{c\}$ | 2.691 | 1.136 | 0.667 | 21.778 | 0.907 |
| $\{a,b,c\}$ | 1.834 | 1.253 | 0.670 | 21.917 | 0.913 |

Table 1: Example with $\mu_e = 10, \sigma^2 = 1$ example with three players of size $n_a = 1, n_b = 8, n_c = 15$. Note that $\{a,b\},\{c\}$ minimizes total cost, but is not individually stable: player $a$ wishes to leave its coalition to join player $c$, which welcomes that player joining it. This produces $\{a,c\},\{b\}$, which is the only individually stable arrangement, giving a Price of Anarchy value of $21.875/21.778 = 1.0045$.

**Lemma 9** (Merging). *Consider two groups of players, $P, Q$. First, merge together the two groups to form $P \cup Q$. Then, remove players from $P \cup Q$ to local learning, removing them in descending order of size. Stop removing players when the first player would prefer to stay (removing it would increase its error). Then, this overall process maintains or decreases total error. In other words,*

$$f_w(Q) + f_w(P) \geq f_w(\{Q \cup P\} \setminus L) + \sum_{i \in L} f_w(\{n_i\}) \tag{3}$$

*where $L$ is the set of large players removed in descending order of size. The inequality is strict so long as the final structure is not identical to the first, up to renaming of players, and it is* not *the case that all the players have the exact same size.*

The proof of Theorem 1 is given simply by applying the lemmas sequentially to show that any other partition $\Pi'$ can be converted to the described partition $\Pi$ through a series of operations that decrease or hold constant total cost.

## 5 Price of Anarchy

The previous section defined the "optimality" of a federating arrangement as its average error, and additionally provided an efficient algorithm to calculate a lowest-cost arrangement. Given that much of prior work (Donahue and Kleinberg [2021], Hasan [2021]) has studied the stability of cooperative games induced by federated learning, the next natural question is to study the relationship between stability and optimality. This section analyzes this relationship, using the canonical game theoretic tools of Price of Anarchy and Price of Stability. All proofs for this section are in Appendix C.

First, we will define the notions of stability under analysis, which are all drawn from standard cooperative game theory literature (Bogomolnaia and Jackson [2002]). A partition of players $\Pi$ is *core stable* if there does not exist a set of players that all would prefer leave their location in $\Pi$ and form a coalition together. A partition is *individually stable* (IS) if there does not exist a single player $i$ that wishes to join some existing coalition $C$, where all members of $C$ weakly prefer that $i$ join. Our results will primarily use the notion of individual stability.

As a reminder, the Price of Anarchy (PoA) is the ratio between the worst (highest-cost) stable arrangement and the best (lowest-cost) arrangement. The Price of Stability is the ratio of the best stable arrangement and the best overall arrangement (regardless of if it is stable or not) (Anshelevich et al. [2008]). Note that the Price of Stability is 1 when there exists an optimal arrangement that is also stable.

First, we will show that for certain ranges of parameter space, the Price of Anarchy and/or Price of Stability are equal to 1. Specifically, Lemma 10 shows that when all players have relatively few samples (no more than $\frac{\mu_e}{\sigma^2}$ each), the grand coalition $\pi_g$ is core stable, implying a Price of (Core) Stability of 1. Recall that $\mu_e$ and $\sigma^2$ are parameters of the federated learning model reflecting the average noise of the data and the average dissimilarity between federating agents, respectively.

**Lemma 10.** *For a set of players with $n_i \leq \frac{\mu_e}{\sigma^2} \; \forall i$, the grand coalition $\pi_g$ is always core stable.*

On the other hand, Lemma 11 shows that when all players have relatively many samples (at least $\frac{\mu_e}{\sigma^2}$ each), every core or individually stable arrangement is also optimal, which means that the Price of Anarchy for this situation is 1.

**Lemma 11.** *For a set of players with $n_i \geq \frac{\mu_e}{\sigma^2} \forall i$, any arrangement that is core stable or individually stable is also optimal.*

However, it is *not* the case that either the Price of Stability or Price of Anarchy is always 1. Table 1 contains an example demonstrating this: there exists a simple three-player case where the optimal arrangement is not individually stable. However, the Price of Anarchy value here is quite small, which suggests the prospect that the Price of Anarchy in general could be bounded.

The main result of this section is Theorem 2, which proves a Price of Anarchy bound of 9 for this problem: the cost of the highest stable arrangement is no more than 9 times the cost of the optimal (lowest cost) arrangement.

**Theorem 2** (Price of Anarchy). *Denote $\Pi_M$ to be a maximum-cost individually stable (IS) partition and $\Pi_{opt}$ to be an optimal (lowest-cost) partition. Then,*

$$PoA = \frac{f_w(\Pi_M)}{f_w(\Pi_{opt})} \leq 9$$

In Theorem 2, the numerator is the cost of $\Pi_M$, a maximum-cost partition, and the denominator is $\Pi_{opt}$, an optimal (lowest-cost) partition. Recall that Definition 1 gives the cost of an arrangement as the weighted sum of the errors of the respective players. Therefore, to get an upper bound on the Price of Anarchy, we will upper bound the errors players experience in $\Pi_M$ and lower bound on the error players experience in $\Pi_{opt}$.

**Summary of proof technique**   Again, this section will show how the larger theorem is the result of several lemmas that act as building blocks. In particular, the lemmas will take two separate approaches towards creating the bound. Lemmas of the first type (12, 13, 14) all provide upper or lower bounds on the errors certain players can experience. These conditions depend on the size of the player (how many samples it has) and the size of the group it is federating with (how many samples in total the rest of the coalition has). For example, Lemmas 12 and 13 taken together show that a player with at least $\frac{\mu_e + \sigma^2}{2\sigma^2}$ samples has a worst-case error no more than 2 times its best-case error. The same pair of lemmas give a multiplicative bound of 9 for players with numbers of samples that falls between $\frac{\mu_e}{9 \cdot \sigma^2}$ and $\frac{\mu_e + \sigma^2}{2\sigma^2}$. Finally, Lemmas 14 and 13 together give a factor of 7.5 for players with fewer than $\frac{\mu_e}{9 \cdot \sigma^2}$ samples that are federating with other players of total size at least $\frac{\mu_e}{3 \cdot \sigma^2}$. Taken together, these errors show that, for almost all cases, the highest error a player experiences is no more than 9 times higher than the lowest error it might experience.

The final case that needs to be addressed is when a player of size $\leq \frac{\mu_e}{9 \cdot \sigma^2}$ is federating in a group with other players of total size $\leq \frac{\mu_e}{3 \cdot \sigma^2}$. Lemma 15 handles this last case by an argument around stability. Specifically, it shows that any players in such an arrangement can only be stable if all of them are grouped together into a single federating coalition. In the proof of Theorem 2, this result ends up enabling an additive factor to the Price of Anarchy bound, which is absorbed into the other factors for a total Price of Anarchy value of 9.

**Statement and description of supporting lemmas**   Next, we will walk through each lemma specifically. Lemma 12 gives an *upper* bound of $\frac{\mu_e}{n_i}$ on the error any player experiences in $\Pi_M$.

**Lemma 12.** *If $\Pi_M$ is a maximum-cost IS partition, then $err_i(\Pi_M) \leq \frac{\mu_e}{n_i}$ for all players $i$.*

*Proof.* Because $\Pi_M$ is individually stable, every player must get error no more than the error it would receive alone (doing local learning). By Lemma 1 with $C = n_i$, a player with samples $n_i$ player gets error $\frac{\mu_e}{n_i}$ alone. □

Next, Lemma 13 provides *lower* bounds on the error a player can receive in $\Pi_{opt}$. It does this by bounding the minimum error a player could get in any arrangement. Again, because the cost of $\Pi_{opt}$ is simply the weighted sum of errors of each individual player, this helps to upper bound the Price of Anarchy. First, Lemma 13 shows that for players with at least $\frac{\mu_e + \sigma^2}{2\sigma^2}$ samples, the lowest possible error it could experience is $\frac{1}{2} \cdot \frac{\mu_e}{n_j}$, which is a factor of 2 off from its worst-case error in Lemma 12. For players with fewer samples than $\frac{\mu_e + \sigma^2}{2\sigma^2}$, Lemma 13 says that the lowest error a player

could experience is $\sigma^2$. This means that the ratio between the two errors is lower than 9 so long as $n_j \geq \frac{\mu_e}{9 \cdot \sigma^2}$. Therefore, in order to get a factor of 9 bound for the overall Price of Anarchy, we need to handle the case of players with size $\leq \frac{\mu_e}{9 \cdot \sigma^2}$, when players have very few samples.

**Lemma 13.** *Consider a player $n_j$ and any set of players $C$. Then, we can lower bound the error player $j$ recieves by federating with $C$:*

$$err_j(C \cup \{n_j\}) \geq \begin{cases} \frac{1}{2} \cdot \frac{\mu_e}{n_j} & n_j \geq \frac{\mu_e + \sigma^2}{2\sigma^2} \\ \sigma^2 & otherwise \end{cases}$$

Lemma 14 is the first of two lemmas handling the case of players with very few samples. It shows that, if a player of size $\leq \frac{\mu_e}{3 \cdot \sigma^2}$ is federating with a set of players of total size at least $\frac{\mu_e}{3 \cdot \sigma^2}$, it is possible to *upper* bound on the error of players in $\Pi_M$ by $7.5 \cdot \sigma^2$. Given the lower bound of $\sigma^2$ in Lemma 13, these together show that there is a ratio of 7.5 at most between the error this player experiences in its best and worst-case arrangements.

**Lemma 14.** *Consider a player $j$ federating with a coalition $C$. If the total number of samples $N_C$ is at least $\frac{\mu_e}{3\sigma^2}$, then $err_j(C \cup \{n_j\}) \leq 7.25 \cdot \sigma^2$.*

However, Lemma 14 does not handle one situation: what if a player of size $\leq \frac{\mu_e}{9 \cdot \sigma^2}$ is federating with a group of players of total size $\leq \frac{\mu_e}{3 \cdot \sigma^2}$? Lemma 15 addresses this last case: it shows that the only such arrangement that is stable is one where all such players are grouped together into a single arrangement. Note that this lemma is itself should not be obvious: it is composed of multiple sub-lemmas which are stated and proved in the appendix. The fact that there can be only one group of such players is used in the Theorem 2 to create an overall bound of 9.

**Lemma 15.** *Consider an arrangement of players, all of size $\leq \frac{\mu_e}{3\sigma^2}$, where at least one player is in a federating cluster where the total mass of its partners is no more than $\frac{\mu_e}{3\sigma^2}$. Then, the only stable arrangement of these players is to have all of them federating together.*

The full proof of Theorem 2 uses these lemmas collectively in order to get an overall Price of Anarchy bound of 9, showing that the worst individually stable arrangement has total cost no more than 9 times the optimal cost.

# 6 Conclusion

In this work, we have given the first Price of Anarchy bound for a game-theoretic model of federated learning. This bound quantifies a key tension between individual incentives and overall societal goals, answering a key question left open in prior literature. Beyond this bound, we also provide an efficient algorithm to calculate an optimal partition of players into federating coalitions, and have characterized conditions where the Price of Anarchy and/or Price of Stability is equal to 1.

There are multiple fascinating extensions to this work. To begin with, other definitions of societal cost (for example, weighting players' errors differently) could produce different Price of Anarchy bounds. Additionally, further work could model more sophisticated methods of federation, including models of domain adaptation. Finally, it would be interesting to explore other notions of societal interest. For example, one vein of research is fairness: how are error rates divided among federating players? Questions might revolve around the maximum gap in error rates between players and whether players that contribute more samples are always rewarded with lower error. Beyond these avenues, though, we believe that the broad topic of federated learning will continue to contain multiple useful and interesting research directions.

# 7 Ethics and societal impact

Given this work's focus defining notions of optimality, there are important ethical considerations. In particular, "optimality" can be defined in multiple different ways: Section 4 motivates the definition we use and Appendix A discusses the merits of other definitions. In particular, it is worth emphasizing that "optimality" is a technical term in optimization and game theory which is always with respect to a given objective function and does not imply a more holistic notion of how desirable a certain solution is. For example, an arrangement could be "optimal" and still be unfair in how errors are distributed among players.

Although our methodology is application-agnostic, federated learning is a machine learning tool that could be applied towards positive goals (e.g. predicting patient outcomes at hospitals) or negative goals (e.g. used to surveil and control populations). It is also worth considering, for each application, whether there could be some other approach that would better address the need. For example, it may be worth considering whether approaches aiming at increasing the number of samples available for low-resource agents would do a better job of increasing the benefit of a federated learning solution. It may even be the case that a solution beyond machine learning would be preferable, such as interventions to reduce the need for a predictive model.

## Acknowledgments and Disclosure of Funding

This work was supported in part by a Simons Investigator Award, a Vannevar Bush Faculty Fellowship, MURI grant W911NF-19-0217, AFOSR grant FA9550-19-1-0183, ARO grant W911NF19-1-0057, a Simons Collaboration grant, a grant from the MacArthur Foundation, and NSF grant DGE-1650441. We are grateful to the AI, Policy, and Practice working group at Cornell for invaluable discussions.

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
