# A  Alternate definitions of optimality

The definition of cost used in this paper is given in Definition 1, which says that an arrangement is optimal if it minimizes the weighted sum of errors over players. As discussed previously, this definition is well-motivated by existing federated learning literature. Additionally, it matches the societal good perspective when the unit society cares about is at the level of the data point. For example, consider the example when the federating agents are hospitals and data points represent individual patients. Then, society as a whole likely cares about minimizing the overall error patients experience, which corresponds to the per-data-point notion of error.

However, other cost functions are worth discussing. For example, Definition 2 gives an unweighted notion of error:

**Definition 2** (Unweighted cost). *The unweighted cost function is given by summing the error over each of the players, without any weighting with respect to size:*

$$f_u(\Pi) = \sum_{C \in \Pi} f_u(C) = \sum_{C \in \Pi} \sum_{i \in C} err_i(C)$$

This definition might be better in a model where the unit society cares about is at the level of the agent. For example, consider a situation where the individual federating agent is a cell phone owned by a single person and data points are word predictions. Then, society as a whole might care about minimizing the sum of errors that individual cell phone users experience, which is given by the unweighted error function.

Finally, we may wish to consider some completely different weight function, given by the definition below:

**Definition 3** (Arbitrary weights). *The arbitrary cost metric is given by summing the weight over each of the players according to some weight $\sum_{i \in [M]} p_i = 1$*

$$f_a(\Pi) = \sum_{C \in \Pi} f_a(C) = \sum_{C \in \Pi} \sum_{i \in C} p_i \cdot err_i(C)$$

Definitions like this have been analyzed in Li et al. [2019], Mohri et al. [2019], Laguel et al. [2021], Chen et al. [2021]. For example, the set of weights $\{p_i\}$ could have fairness goals, attempting to up-weight players with higher error. Alternatively, it could represent some notion of the data quality players are contributing, with players producing more or lower-error players being weighted more.

In this work, we selected Definition 1 (weighted error) based on its standard use in the federated learning literature. Analysis of the same type (calculating an optimal arrangement and analyzing the Price of Anarchy) could be completed for any other definition of cost, but would require new proofs for calculation of optimal arrangements and for any Price of Anarchy bound.

# B  Optimality calculation

**Lemma 2.** *Consider a partition $\Pi$ made up of coalitions $\{C_i\}$. Then, using the error form given in Equation 2, the total cost of $\Pi$ is given by*

$$f_w(\Pi) = \sum_{C \in \Pi} \left\{ \mu_e + \sigma^2 \cdot N_C - \sigma^2 \frac{\sum_{i \in C} n_i^2}{N_C} \right\}$$

*Proof.*

$$f_w(C) = \sum_{j \in C} err_j(C) \cdot n_i = \sum_{j \in C} \left( \frac{\mu_e}{\sum_{i \in C} n_i} + \sigma^2 \frac{\sum_{i \neq j} n_i^2 + \left( \sum_{i \neq j} n_i \right)^2}{\left( \sum_{i \in C} n_i \right)^2} \right) \cdot n_j$$

$$= \sum_{j \in C} \frac{\mu_e}{\sum_{i \in C} n_i} \cdot n_j + \sigma^2 \sum_{j \in C} \frac{\sum_{i \neq j} n_i^2 + \left( \sum_{i \neq j} n_i \right)^2}{\left( \sum_{i \in C} n_i \right)^2} \cdot n_j$$

$$= \mu_e + \sigma^2 \sum_{j \in C} \frac{n_j \cdot \sum_{i \neq j} n_i^2 + n_j \cdot (N_C - n_j)^2}{N_C^2}$$

where we have used $N_C = \sum_{i \in C} n_i$. Focusing solely on the numerator of the second term, we simplify:

$$\sum_{j \in C} \left\{ n_j \cdot \sum_{i \neq j} n_i^2 + n_j \cdot N_C^2 + n_j^3 - 2N_C \cdot n_j^2 \right\} = \sum_{j \in C} n_j \cdot \sum_{i \in C} n_i^2 + N_C^2 \sum_{j \in C} n_j - 2N_C \sum_{j \in C} n_j^2$$

$$= N_C \cdot \sum_{i \in C} n_i^2 + N_C^3 - 2N_C \sum_{i \in C} n_i^2 = N_C^3 - N_C \cdot \sum_{i \in C} n_i^2$$

Combining this with the rest of the term gives:

$$\mu_e + \sigma^2 \cdot \frac{N_C^3 - N_C \cdot \sum_{i \in C} n_i^2}{N_C^2} = \mu_e + \sigma^2 \cdot N_C - \sigma^2 \frac{\sum_{i \in C} n_i^2}{N_C}$$

$\square$

**Lemma 3.** $\forall \rho > 1$, *there exists a setting where local learning results in average error more than $\rho$ times higher than optimal:* $\frac{f_w(\pi_l)}{f_w(OPT)} > \rho$.

*Proof.* We will prove this result by the setting where $M$ players each have $n$ samples, for $M > \rho$ and any $\mu_e, \sigma^2, n \in \mathbb{N}_{\geq 1}$ such that $n < \left( \frac{M}{\rho} - 1 \right) \frac{\mu_e}{(M-1) \cdot \sigma^2}$.

In this simplified setting where all of the players have the same number of samples, the cost of a coalition $C$ involving $M$ players is given by:

$$\mu_e + \sigma^2 \cdot n \cdot M - \sigma^2 \cdot \frac{M \cdot n^2}{M \cdot n} = \mu_e + \sigma^2 \cdot n \cdot (M - 1)$$

For our given example, $n < \frac{\mu_e}{\sigma^2}$, which implies that "merging" any two groups $A$ and $B$ will reduce total cost:

$$f_w(A) + f_w(B) = \mu_e + \sigma^2 \cdot n \cdot (M_A - 1) + \mu_e + \sigma^2 \cdot n \cdot (M_B - 1)$$
$$> \mu_e + \sigma^2 \cdot n \cdot (M_A + M_B - 1)$$
$$= f_w(A \cup B)$$

This implies that the optimal cost is achieved by $\pi_g$, given by $\mu_e + \sigma^2 \cdot (M - 1)$. Conversely, the cost of having $M$ players doing local learning is:

$$f_w(\pi_l) = \sum_{i=1}^{M} \left\{ \mu_e + \sigma^2 \cdot n - \sigma^2 \cdot \frac{n^2}{n} \right\} = \sum_{i=1}^{M} \mu_e = \mu_e \cdot M$$

Combining these facts gives:

$$\frac{f_w(\pi_l)}{f_w(OPT)} = \frac{\mu_e \cdot M}{\mu_e + \sigma^2 \cdot (M - 1) \cdot n} = \frac{M}{1 + \frac{\sigma^2}{\mu_e} \cdot (M - 1) \cdot n}$$

$$> \frac{M}{1 + \frac{\sigma^2}{\mu_e} \cdot (M - 1) \cdot \frac{\mu_e}{(M-1) \cdot \sigma^2} \cdot \left( \frac{M}{\rho} - 1 \right)}$$

$$= \frac{M}{1 + \frac{M}{\rho} - 1} = \rho$$

as desired.

$\square$

**Lemma 4.** $\forall \rho > 1$, *there exists a setting where federating in the grand coalition results in average error more than $\rho$ times higher than optimal:* $\frac{f_w(\pi_g)}{f_w(OPT)} > \rho$.

*Proof.* We will prove this result by the setting where $M$ players each have $n$ samples, with $M > \frac{1}{\rho}$ and any $\mu_e, \sigma^2, n \in \mathbb{N}_{\geq 1}$ such that $n > \max\left[\frac{\mu_e}{\sigma^2 \cdot (M-1)} \cdot (\rho \cdot M - 1), \frac{\mu_e}{\sigma^2}\right]$.

The initial construction follows similarly to Lemma 3. For our given example, $n > \frac{\mu_e}{\sigma^2}$, which implies that "merging" any two groups $A$ and $B$ will *increase* total cost:

$$f_w(A) + f_w(B) = \mu_e + \sigma^2 \cdot n \cdot (M_A - 1) + \mu_e + \sigma^2 \cdot n \cdot (M_B - 1)$$
$$< \mu_e + \sigma^2 \cdot n \cdot (M_A + M_B - 1)$$
$$= f_w(A \cup B)$$

This implies that the optimal cost is achieved $\pi_l$. Using the value derived in the proof of Lemma 3, we have:

$$\frac{f_w(\pi_g)}{f_w(OPT)} = \frac{\mu_e + \sigma^2 \cdot (M-1) \cdot n}{\mu_e \cdot M}$$
$$= \frac{1 + \frac{\sigma^2}{\mu_e} \cdot (M-1) \cdot n}{M}$$
$$> \frac{1 + \frac{\sigma^2}{\mu_e} \cdot (M-1) \cdot \max\left[\frac{\mu_e}{\sigma^2 \cdot (M-1)} \cdot (\rho \cdot M - 1), \frac{\mu_e}{\sigma^2}\right]}{M}$$
$$\geq \frac{1 + \frac{\sigma^2}{\mu_e} \cdot (M-1) \cdot \frac{\mu_e}{\sigma^2 \cdot (M-1)} \cdot (\rho \cdot M - 1)}{M}$$
$$= \frac{1 + \rho \cdot M - 1}{M} = \rho$$

as desired. $\qquad\qquad\square$

The proof of Theorem 1, below, relies on multiple sub-lemmas which are stated and proved immediately afterwards.

**Theorem 1.** *Consider a set of players $\{n_i\}$. An optimal partition $\Pi$ can be created as follows: first, start with every player doing local learning. Then, begin by grouping the players together in ascending order of size, stopping when the first player would increase its error by joining the coalition from local learning. Then, the resulting partition $\Pi$ is optimal.*

*Proof.* First, we note two special cases. If $\{n_i\} \leq \frac{\mu_e}{\sigma^2}$, then by Lemma 10 (stated and proved later in this appendix) the grand coalition $\pi_g$ is core stable. For the grand coalition, core stability implies individual stability, so we know that every player prefers $\pi_g$ to local learning. This implies that, following the steps given in this theorem, every player will prefer to join the growing coalition as opposed to doing local learning, and so the optimal arrangement is $\pi_g$.

Next, if $\{n_i\} > \frac{\mu_e}{\sigma^2}$, then by Lemma 5.3 in Donahue and Kleinberg [2021] every player minimizes their error in $\pi_l$ (local learning). As a result, using the algorithm given in the statement of this theorem, every player will increase their error by combining with another player, so $\pi_l$ is optimal. If $\{n_i\} \geq \frac{\mu_e}{\sigma^2}$ (some players have exactly $\frac{\mu_e}{\sigma^2}$ samples), then all players with $n_i = \frac{\mu_e}{\sigma^2}$ will be indifferent towards being merged with any other player also of size $\frac{\mu_e}{\sigma^2}$, but no player of size strictly greater than $\frac{\mu_e}{\sigma^2}$ will be able to be merged. The resulting optimal arrangement will have all of the players of size exactly $\frac{\mu_e}{\sigma^2}$ together, with all other players doing local learning, and will have cost identical to $\pi_l$.

Finally, we will consider the case where some players have size strictly less than $\frac{\mu_e}{\sigma^2}$ and some have strictly more. Call the partition calculated by following the steps of this theorem $\Pi$, and consider any other coalition partition $\Pi'$. We will convert $\Pi'$ into $\Pi$ using only cost reducing or maintaining steps, which will show that $\Pi$ is optimal. We will refer to players with size $\leq \frac{\mu_e}{\sigma^2}$ as small, and players of size $> \frac{\mu_e}{\sigma^2}$ as large.

- If there are any coalitions where players would prefer to leave the coalition, remove them in order of descending size. Note: a coalition made up of only players of size smaller than $\frac{\mu_e}{\sigma^2}$ will never have players leave. A coalition made up of only players of size larger than $\frac{\mu_e}{\sigma^2}$ will always wish to have players leave. This reduces total cost by Lemma 5.

- Every coalition of size 2 or larger will have at least one small player in it. Begin merging all such coalitions (as well as any small players doing local learning), removing large players as necessary (in descending size, if they would prefer local learning). Note that the merging operation will never remove a small player, so it always strictly reduces the number of coalitions involving small players. This reduces cost by Lemma 9.

- When all of the small players are in one coalition, if there are large players in the coalition as well, check if they are the smallest possible large player. If not, swap them for smaller large players iteratively (ones that are doing local learning) until the players in the coalition are doing local learning. By Lemma 6, this reduces cost.

- Add large players in increasing order of size (if any wish to join). From Lemma 7 we know that if player $n_i$ doesn't wish to join a coalition, then neither will any player of size $n_j \geq n_i$. From Lemma 5, adding any player that wishes to join reduces total cost.

- If no players wish to join, then remove large players in descending order of size if they would prefer local learning, which again from Lemma 5 would reduce cost. From Lemma 8, if a player of size $n_i$ doesn't wish to leave, then all other players of size $n_j \leq n_i$ also do not wish to leave.

The final arrangement exactly matches $\Pi$. $\qquad\square$

**Lemma 5** (Equivalence of player preference and reducing cost). *Take any coalition $Q$ and any player $j$. Then, a player wishes to join that coalition (from local learning) if and only if doing so would reduce total cost. That is,*

$$f_w(\{n_j\}) + f_w(Q) \geq f_w(\{n_j\} \cup Q) \quad \Leftrightarrow \quad err_j(\{n_j\}) \geq err_j(\{n_j\} \cup Q)$$

*Proof.* This proof will work by showing the forms of the inequalities are identical. We will start with the cost inequality:

$$f_w(\{n_j\}) + f_w(Q) \geq f_w(\{n_j\} \cup Q)$$

$$\mu_e + \mu_e + \sigma^2 \cdot N_Q - \sigma^2 \cdot \frac{\sum_{i \in Q} n_i^2}{N_Q} \geq \mu_e + \sigma^2 \cdot N_Q + \sigma^2 \cdot n_j - \sigma^2 \frac{\sum_{i \in Q} n_i^2 + n_j^2}{N_Q + n_j}$$

$$\mu_e \geq \sigma^2 \cdot n_j - \sigma^2 \frac{\sum_{i \in Q} n_i^2 + n_j^2}{N_Q + n_j} + \sigma^2 \cdot \frac{\sum_{i \in Q} n_i^2}{N_Q}$$

Bringing all terms over common denominator on the righthand side:

$$\mu_e \geq \sigma^2 \frac{n_j \cdot (N_Q + n_j) \cdot N_Q - N_Q \sum_{i \in Q} n_i^2 - n_j^2 \cdot N_Q + N_Q \sum_{i \in Q} n_i^2 + n_j \cdot \sum_{i \in Q} n_i^2}{(N_Q + n_j) \cdot N_Q}$$

$$\mu_e \geq \sigma^2 \frac{n_j \cdot (N_Q + n_j) \cdot N_Q - n_j^2 \cdot N_Q + n_j \cdot \sum_{i \in Q} n_i^2}{(N_Q + n_j) \cdot N_Q}$$

$$\mu_e \geq \sigma^2 \frac{n_j \cdot N_Q^2 + n_j^2 \cdot N_Q - n_j^2 \cdot N_Q + n_j \cdot \sum_{i \in Q} n_i^2}{(N_Q + n_j) \cdot N_Q}$$

$$\mu_e \geq \sigma^2 \frac{n_j \cdot N_Q^2 + n_j \cdot \sum_{i \in Q} n_i^2}{(N_Q + n_j) \cdot N_Q}$$

$$\mu_e \geq \sigma^2 \cdot \frac{n_j}{N_Q + n_j} \cdot \frac{N_Q^2 + \sum_{i \in Q} n_i^2}{N_Q}$$

Next, we will reduce the error inequality to the same form:

$$err_j(\{n_j\}) \geq err_j(\{n_j\} \cup Q)$$

$$\frac{\mu_e}{n_j} \geq \frac{\mu_e}{N_Q + n_j} + \sigma^2 \cdot \frac{\sum_{i \in Q} n_i^2 + N_Q^2}{(N_Q + n_j)^2}$$

$$\frac{\mu_e}{n_j} - \frac{\mu_e}{N_Q + n_j} \geq \sigma^2 \cdot \frac{\sum_{i \in Q} n_i^2 + N_Q^2}{(N_Q + n_j)^2}$$

$$\mu_e \cdot \frac{N_Q}{n_j \cdot (N_Q + n_j)} \geq \sigma^2 \cdot \frac{\sum_{i \in Q} n_i^2 + N_Q^2}{(N_Q + n_j)^2}$$

$$\mu_e \geq \sigma^2 \cdot \frac{(N_Q + n_j) \cdot n_j}{N_Q} \frac{\sum_{i \in Q} n_i^2 + N_Q^2}{(N_Q + n_j)^2}$$

$$\mu_e \geq \sigma^2 \cdot \frac{n_j}{N_Q + n_j} \cdot \frac{N_Q^2 + \sum_{i \in Q} n_i^2}{N_Q}$$

as desired. $\qquad\square$

**Lemma 6** (Swapping). *Take any set $Q$ including a player $n_j > n_k$, where the player $n_k$ is doing local learning. Then, swapping the roles of players $k$ and $j$ always decreases total cost.*

$$f_w(Q \cup \{n_j\}) + f_w(\{n_k\}) > f_w(Q \cup \{n_k\}) + f_w(\{n_j\})$$

*Proof.* We write out each side:

$$\mu_e + \sigma^2 \cdot N_Q + \sigma^2 \cdot n_j - \sigma^2 \frac{\sum_{i \in Q} n_i^2 + n_j^2}{N_Q + n_j} + \mu_e > \mu_e + \sigma^2 \cdot N_Q + \sigma^2 n_k - \sigma^2 \frac{\sum_{i \in Q} n_i^2 + n_k^2}{N_Q + n_k} + \mu_e$$

$$\sigma^2 \cdot n_j - \sigma^2 \cdot \frac{\sum_{i \in Q} n_i^2 + n_j^2}{N_Q + n_j} > \sigma^2 \cdot n_k - \sigma^2 \cdot \frac{\sum_{i \in Q} n_i^2 + n_k^2}{N_Q + n_k}$$

Dropping the common $\sigma^2$ term for clarity:

$$n_j - \frac{\sum_{i \in Q} n_i^2 + n_j^2}{N_Q + n_j} > n_k - \frac{\sum_{i \in Q} n_i^2 + n_k^2}{N_Q + n_k}$$

In order to prove the above inequality, we will consider the following fraction:

$$x - \frac{\sum_{i \in Q} n_i^2 + x^2}{N_Q + x}$$

and want to show this is increasing with respect to $x$. The derivative of the function gives:

$$1 - \frac{2x \cdot (N_Q + x) - (\sum_{i \in Q} n_i^2 + x^2) \cdot 1}{(N_Q + x)^2}$$

$$= \frac{1}{(N_Q + x)^2} \cdot \left( N_Q^2 + x^2 + 2x \cdot N_Q - \left( 2x \cdot (N_Q + x) - \left( \sum_{i \in Q} n_i^2 + x^2 \right) \right) \right)$$

$$= \frac{1}{(N_Q + x)^2} \cdot \left( N_Q^2 + x^2 + 2x \cdot N_Q - 2x \cdot N_Q - 2x^2 + \left( \sum_{i \in Q} n_i^2 + x^2 \right) \right)$$

$$= \frac{1}{(N_Q + x)^2} \cdot \left( N_Q^2 + \sum_{i \in Q} n_i^2 \right)$$

which is positive, as desired. This implies that the original inequality is satisfied, meaning that the swapping of the roles of players $j, k$ decreases total cost. $\qquad\square$

**Lemma 7** (Monotonicity of joining). *If a player of size $n_j$ would prefer local learning to joining a coalition $Q$, then any player of size $n_k \geq n_j$ also prefers local learning to joining the same coalition. That is, for $n_k \geq n_j$,*

$$err_j(Q \cup \{n_j\}) \geq err_j(\{n_j\}) \quad \Rightarrow \quad err_k(Q \cup \{n_k\}) \geq err_k(\{n_k\})$$

*Conversely, if a player $j$ wishes to join $Q$, then any other player of size $n_k \leq n_j$ would have also wanted to join. That is, for $n_j \geq n_k$,*

$$err_j(Q \cup \{n_j\}) \leq err_j(\{n_j\}) \quad \Rightarrow \quad err_k(Q \cup \{n_k\}) \leq err_k(\{n_k\})$$

*Proof.* The initial premise depends on whether or not the below inequality is satisfied:

$$err_j(Q \cup \{n_j\}) \geq err_j(\{n_j\})$$

$$\frac{\mu_e}{N_Q + n_j} + \sigma^2 \frac{\sum_{i \in Q} n_i^2 + \left(\sum_{i \in Q} n_i\right)^2}{(N_Q + n_j)^2} \geq \frac{\mu_e}{n_j}$$

Rearranging:

$$\sigma^2 \frac{\sum_{i \in Q} n_i^2 + \left(\sum_{i \in Q} n_i\right)^2}{(N_Q + n_j)^2} \geq \frac{\mu_e}{n_j} - \frac{\mu_e}{N_Q + n_j}$$

$$\sigma^2 \left(\sum_{i \in Q} n_i^2 + \left(\sum_{i \in Q} n_i\right)^2\right) \geq (N_Q + n_j)^2 \cdot \left(\frac{\mu_e}{n_j} - \frac{\mu_e}{N_Q + n_j}\right)$$

$$\sigma^2 \left(\sum_{i \in Q} n_i^2 + \left(\sum_{i \in Q} n_i\right)^2\right) \geq (N_Q + n_j)^2 \cdot \frac{\mu_e \cdot N_Q}{n_j \cdot (N_Q + n_j)}$$

$$\sigma^2 \left(\sum_{i \in Q} n_i^2 + \left(\sum_{i \in Q} n_i\right)^2\right) \geq (N_Q + n_j) \cdot \frac{\mu_e \cdot N_Q}{n_j}$$

$$\sigma^2 \left(\sum_{i \in Q} n_i^2 + \left(\sum_{i \in Q} n_i\right)^2\right) \geq \mu_e \cdot \frac{N_Q^2}{n_j} + \mu_e \cdot N_Q$$

The lefthand side is a constant independent of $n_k$ and the righthand side is a constant plus a term that is decreasing in $n_j$. If the original inequality ($err_j(Q \cup \{n_j\}) \geq err_j(\{n_j\})$) is satisfied, then it will also be satisfied for any $n_k \geq n_j$ (implying $err_k(Q \cup \{n_k\}) \geq err_k(\{n_k\})$). Conversely, if the original inequality is not satisfied (so $err_j(Q \cup \{n_j\}) \leq err_j(\{n_j\})$), then it will also not be satisfied for any $n_k \leq n_j$ (implying $err_k(Q \cup \{n_k\}) \leq err_k(\{n_k\})$). □

**Lemma 8** (Monotonicity of leaving). *Take any coalition $Q$. Then, if any player $j \in Q$ of size $n_j$ wishes to leave $Q$ for local learning, then any player of size $n_k \geq n_j$ also wishes to leave for local learning. That is, for $n_k \geq n_j$*

$$err_j(Q) \geq err_j(\{n_j\}) \quad \Rightarrow \quad err_k(Q) \geq err_k(\{n_k\})$$

*Conversely, if a player $j \in Q$ of size $n_j$ does not wish to leave $Q$ for local learning, then any player $k \in Q$ of size $n_k \leq n_j$ also does not wish to leave. That is, for $n_k \leq n_j$*

$$err_j(Q) \leq err_j(\{n_j\}) \quad \Rightarrow \quad err_k(Q) \leq err_k(\{n_k\})$$

*Proof.* First, we will prove the first statement. Suppose by contradiction that some $n_j$ wishes to leave, but another player of size $n_k \geq n_j$ does not wish to. First, we remove $n_j$ for local learning, which by Lemma 5 reduces total cost. Next, we swap the role of players $j$ and $k$, which by Lemma 6 again reduces or keeps constant total cost. We have constructed a series of operations that either reduce or

keep constant total cost, and results in an arrangement equivalent to simply removing player $j$. By Lemma 5, this means that player $j$ originally would have wished to leave.

Next, we will prove the second statement. Suppose by contradiction that some player $k$ wishes to leave, even though another player $n_j > n_k$ does not wish to leave. First, we remove player $k$ to local learning: if it wishes to leave, then by Lemma 5 removing it reduces or keeps constant total cost. Then, by Lemma 6, we can reduce total cost by swapping it with the $n_j > n_k$ player. We have constructed a series of operations that either reduce or keep constant total cost, and results in an arrangement equivalent to simply removing player $j$. But this is exactly equivalent to just removing the $n_j$ player, which we know from Lemma 5 must not reduce total cost (or else player $j$ would wish to leave). $\qquad\square$

**Lemma 9** (Merging). *Consider two groups of players, $P, Q$. First, merge together the two groups to form $P \cup Q$. Then, remove players from $P \cup Q$ to local learning, removing them in descending order of size. Stop removing players when the first player would prefer to stay (removing it would increase its error). Then, this overall process maintains or decreases total error. In other words,*

$$f_w(Q) + f_w(P) \geq f_w(\{Q \cup P\} \setminus L) + \sum_{i \in L} f_w(\{n_i\}) \tag{3}$$

*where $L$ is the set of large players removed in descending order of size. The inequality is strict so long as the final structure is not identical to the first, up to renaming of players, and it is* not *the case that all the players have the exact same size.*

*Proof.* First, we have to reason about what $L$ could be. We will say that player $j$ with $n_j$ samples is a largest element in $P \cup Q$, and WLOG $j \in P$. (If multiple players have $n_j$ samples, it suffices to select one at random.). We will show that, in order to show Equation 3, it suffices to show that:

$$f_w(Q) + f_w(P) \geq f_w(Q \cup P \setminus n_j) + f_w(\{n_j\}) \tag{4}$$

First, assume $L$ is empty. Then, every player wishes to stay in the final group. Then, Equation 3 becomes:

$$f_w(Q) + f_w(P) > f_w(Q \cup P)$$

From Lemma 5, we know that because player $n_j$ doesn't wish to leave $Q \cup P$, removing it must increase total cost:

$$f_w(Q \cup P \setminus \{n_j\}) + f_w(\{n_j\}) > f_w(Q \cup P)$$

So, if we show that Equation 4 is satisfied, then this implies that Equation 3 is satisfied.

Next, we will assume $L = \{n_j\}$. Then, the statement we are trying to show is exactly Equation 4. Finally, let's assume that $|L| \geq 2$: $n_j$ is removed, but so are some others. Again, by Lemma 5, because these players prefer local learning to federation, adding them back in to the coalition increases cost, so

$$f_w(Q) + f_w(P \cup \{n_j\}) > f_w(Q \cup P \setminus n_j) + f_w(\{n_j\})$$

So, it suffices to consider Equation 4: if we prove that this is satisfied, it always implies that Equation 3 is satisfied.

Next, we will prove this statement:

$$f_w(Q) + f_w(P) \geq f_w(Q \cup P \setminus n_j) + f_w(\{n_j\})$$

Plugging in for the form of $f_w(\cdot)$ gives:

$$\mu_e + \sigma^2 \cdot N_Q - \sigma^2 \cdot \frac{\sum_{i \in Q} n_i^2}{N_Q} + \mu_e + \sigma^2 \cdot N_P - \sigma^2 \cdot \frac{\sum_{i \in P} n_i^2}{N_P}$$

$$\geq \mu_e + \sigma^2 \cdot N_Q + \sigma^2 \cdot (N_Q - n_j) - \sigma^2 \frac{\sum_{i \in Q} n_i^2 + \sum_{i \in P} n_i^2 - n_j^2}{N_Q + N_P - n_j} + \mu_e$$

Simplifying gives:

$$\sigma^2 \cdot n_j - \sigma^2 \cdot \frac{\sum_{i \in Q} n_i^2}{N_Q} - \sigma^2 \cdot \frac{\sum_{i \in P} n_i^2}{N_P} \geq -\sigma^2 \frac{\sum_{i \in Q} n_i^2 + \sum_{i \in P} n_i^2 - n_j^2}{N_Q + N_P - n_j}$$

For convenience, we'll drop the common $\sigma^2$ coefficient as we continue simplifying:

$$n_j - \frac{\sum_{i \in Q} n_i^2}{N_Q} - \frac{\sum_{i \in P} n_i^2}{N_P} \geq -\frac{\sum_{i \in Q} n_i^2 + \sum_{i \in P} n_i^2 - n_j^2}{N_Q + N_P - n_j}$$

$$n_j \geq \frac{\sum_{i \in Q} n_i^2}{N_Q} + \frac{\sum_{i \in P} n_i^2}{N_P} - \frac{\sum_{i \in Q} n_i^2 + \sum_{i \in P} n_i^2}{N_Q + N_P - n_j} + \frac{n_j^2}{N_Q + N_P - n_j}$$

$$n_j - \frac{n_j^2}{N_Q + N_P - n_j} \geq \frac{\sum_{i \in Q} n_i^2}{N_Q} + \frac{\sum_{i \in P} n_i^2}{N_P} - \frac{\sum_{i \in Q} n_i^2 + \sum_{i \in P} n_i^2}{N_Q + N_P - n_j}$$

$$n_j \cdot \frac{N_Q + N_P - n_j - n_j}{N_Q + N_P - n_j} \geq \left( \sum_{i \in Q} n_i^2 \right) \cdot \left( \frac{1}{N_Q} - \frac{1}{N_Q + N_P - n_j} \right) + \left( \sum_{i \in P} n_i^2 \right) \cdot \left( \frac{1}{N_P} - \frac{1}{N_Q + N_P - n_j} \right)$$

$$n_j \cdot \frac{N_Q + N_P - 2n_j}{N_Q + N_P - n_j} \geq \left( \sum_{i \in Q} n_i^2 \right) \cdot \frac{N_P - n_j}{N_Q \cdot (N_Q + N_P - n_j)} + \left( \sum_{i \in P} n_i^2 \right) \cdot \frac{N_Q - n_j}{N_P \cdot (N_Q + N_P - n_j)}$$

$$n_j \cdot (N_Q + N_P - 2n_j) \geq \left( \sum_{i \in Q} n_i^2 \right) \cdot \frac{N_P - n_j}{N_Q} + \left( \sum_{i \in P} n_i^2 \right) \cdot \frac{N_Q - n_j}{N_P}$$

$$N_Q + N_P - 2n_j \geq \left( \sum_{i \in Q} \frac{n_i^2}{n_j} \right) \cdot \frac{N_P - n_j}{N_Q} + \left( \sum_{i \in P} \frac{n_i^2}{n_j} \right) \cdot \frac{N_Q - n_j}{N_P}$$

Because $n_j$ is the largest element, we can upper bound each term $\frac{n_i^2}{n_j}$ with $n_i$:

$$N_Q + N_P - 2n_j \geq (N_Q) \cdot \frac{N_P - n_j}{N_Q} + (N_P) \cdot \frac{N_Q - n_j}{N_P}$$

$$N_Q + N_P - 2n_j \geq N_P - N_j + N_Q - n_j$$

This gives an equality, and a strict inequality if $n_i < n_j$ for at least one player. Finally, we note that if the final structure is identical to the original structure, the cost is identical, so the inequality is similarly an equality. $\square$

**Lemma 10.** *For a set of players with $n_i \leq \frac{\mu_e}{\sigma^2} \; \forall i$, the grand coalition $\pi_g$ is always core stable.*

*Proof.* For reference, Donahue and Kleinberg [2021] analyzes a restricted example of $n_i \leq \frac{\mu_e}{\sigma^2}$ case, where players come in two types, $n_s, n_\ell$, both $\leq \frac{\mu_e}{\sigma^2}$. Theorem 6.7 in that work shows that the grand coalition $\pi_g$ is core stable for the two-type case. This lemma extends that result to show that $\pi_g$ is core stable for the broader case of $n_i \leq \frac{\mu_e}{\sigma^2}$, where players may come in more than two sizes.

First, we will assume by contradiction that there exists a set $A \subset C$, where $C$ is the grand coalition, and where we assume that $err_j(A) < err_j(C)$ for every $j \in A$. We will then show that this violates the requirement that $n_i \leq \frac{\mu_e}{\sigma^2}$ for all $i \in C$, indicating that it is impossible for such a coalition $A$ to exist.

By assumption,
$$err_j(C) > err_j(A)$$
Using $N_A = \sum_{i \in A} n_i$ and $N = \sum_{i \in C} n_i$ we have:

$$\frac{\mu_e}{N} + \sigma^2 \cdot \frac{\sum_{i \neq j} n_i^2 + (N - n_j)^2}{N^2} > \frac{\mu_e}{N_A} + \sigma^2 \cdot \frac{\sum_{i \in A, i \neq j} n_i^2 + (N_A - n_j)^2}{N_A^2}$$

Multiplying each side by $n_j$ preserves the inequality:

$$\frac{\mu_e}{N} \cdot n_j + \sigma^2 \cdot \frac{\sum_{i \neq j} n_i^2 + (N - n_j)^2}{N^2} \cdot n_j > \frac{\mu_e}{N_A} \cdot n_j + \sigma^2 \cdot \frac{\sum_{i \in A, i \neq j} n_i^2 + (N_A - n_j)^2}{N_A^2} \cdot n_j$$

Next, we sum each side over all $j \in A$:

$$\sum_{j \in A} \left\{ \frac{\mu_e}{N} \cdot n_j + \sigma^2 \cdot \frac{\sum_{i \neq j} n_i^2 + (N - n_j)^2}{N^2} \cdot n_j \right\} > \sum_{j \in A} \left\{ \frac{\mu_e}{N_A} \cdot n_j + \sigma^2 \cdot \frac{\sum_{i \in A, i \neq j} n_i^2 + (N_A - n_j)^2}{N_A^2} \cdot n_j \right\}$$

We will evaluate this sum term by term. The $\mu_e$ terms are simplest:

$$\sum_{j \in A} \frac{\mu_e}{N} \cdot n_j = \frac{\mu_e}{N} \cdot N_A$$

$$\sum_{j \in A} \frac{\mu_e}{N_A} \cdot n_j = \mu_e$$

For evaluating the sum of the $\sigma^2$ coefficient, we will first note that we can rewrite the numerator:

$$\sum_{i \neq j} n_i^2 + (N - n_j)^2 = \sum_{i \neq j} n_i^2 + N^2 + n_j^2 - 2N \cdot n_j = \sum_{i \in C} n_i^2 + N^2 - 2N \cdot n_j$$

This means that the entire coefficient on the lefthand side can be rewritten as:

$$\sum_{j \in A} \left\{ \sigma^2 \cdot \frac{\sum_{i \neq j} n_i^2 + (N - n_j)^2}{N^2} \cdot n_j \right\} = \sum_{j \in A} \left\{ \sigma^2 \cdot \frac{N^2 + \sum_{i \in C} n_i^2 - 2N \cdot n_j}{N^2} \cdot n_j \right\}$$

$$= \sum_{j \in A} \left\{ \sigma^2 \cdot \left( 1 + \frac{\sum_{i \in C} n_i^2}{N^2} - 2 \frac{n_j}{N} \right) \cdot n_j \right\}$$

$$= \sigma^2 \cdot N_A + \sigma^2 \cdot N_A \frac{\sum_{i \in C} n_i^2}{N^2} - 2 \frac{\sum_{i \in A} n_i^2}{N}$$

Similarly, we can rewrite the numerator of the $\sigma^2$ coefficient on the righthand side:

$$\sum_{i \neq j, i \in A} n_i^2 + (N_A - n_j)^2 = \sum_{i \neq j, i \in A} n_i^2 + N_A^2 + n_j^2 - 2N_A \cdot n_j = \sum_{i \in A} n_i^2 + N_A^2 - 2N_A \cdot n_j$$

Remember that $A \subset C$. Similarly, we can rewrite the entire coefficient as:

$$\sum_{j \in A} \left\{ \sigma^2 \cdot \frac{\sum_{i \neq j, i \in A} n_i^2 + (N_A - n_j)^2}{N_A^2} \cdot n_j \right\} = \sum_{j \in A} \left\{ \sigma^2 \cdot \frac{N_A^2 + \sum_{i \in A} n_i^2 - 2N_A \cdot n_j}{N_A^2} \cdot n_j \right\}$$

$$= \sum_{j \in A} \left\{ \sigma^2 \cdot \left( 1 + \frac{\sum_{i \in A} n_i^2}{N_A^2} - 2 \frac{n_j}{N_A} \right) \cdot n_j \right\}$$

$$= \sigma^2 \cdot N_A + \sigma^2 \cdot N_A \frac{\sum_{i \in A} n_i^2}{N_A^2} - 2 \cdot \sigma^2 \cdot \frac{\sum_{i \in A} n_i^2}{N_A}$$

$$= \sigma^2 \cdot N_A + \sigma^2 \cdot \frac{\sum_{i \in A} n_i^2}{N_A} - 2 \cdot \sigma^2 \cdot \frac{\sum_{i \in A} n_i^2}{N_A}$$

$$= \sigma^2 \cdot N_A - \sigma^2 \cdot \frac{\sum_{i \in A} n_i^2}{N_A}$$

Combining these terms back into the inequality gives:

$$\mu_e \cdot \frac{N_A}{N} + \sigma^2 \cdot N_A + \sigma^2 \cdot \frac{N_A}{N} \cdot \frac{\sum_{i \in C} n_i^2}{N} - 2 \frac{\sum_{i \in A} n_i^2}{N} > \mu_e + \sigma^2 \cdot N_A - \sigma^2 \frac{\sum_{i \in A} n_i^2}{N_A}$$

Simplification:

$$\mu_e \cdot \frac{N_A}{N} + \sigma^2 \cdot \frac{N_A}{N} \cdot \frac{\sum_{i \in C} n_i^2}{N} - 2 \frac{\sum_{i \in A} n_i^2}{N} > \mu_e - \sigma^2 \frac{\sum_{i \in A} n_i^2}{N_A}$$

$$\mu_e \cdot \frac{N_A}{N} + \sigma^2 \cdot \frac{N_A}{N} \cdot \frac{\sum_{i \in C} n_i^2}{N} - \sigma^2 \frac{\sum_{i \in A} n_i^2}{N} > \mu_e - \sigma^2 \frac{\sum_{i \in A} n_i^2}{N_A} + \sigma^2 \frac{\sum_{i \in A} n_i^2}{N}$$

$$\frac{N_A}{N} \cdot \left( \mu_e + \sigma^2 \cdot \frac{\sum_{i \in C} n_i^2}{N} - \sigma^2 \frac{\sum_{i \in A} n_i^2}{N_A} \right) > \mu_e + \sigma^2 \frac{\sum_{i \in A} n_i^2}{N} - \sigma^2 \frac{\sum_{i \in A} n_i^2}{N_A}$$

Note that the terms on the left and the right look very similar. We will strategically add and subtract a term on the left:

$$\frac{N_A}{N} \cdot \left( \mu_e + \sigma^2 \cdot \frac{\sum_{i \in C} n_i^2 - \sum_{i \in A} n_i^2 + \sum_{i \in A} n_i^2}{N} - \sigma^2 \frac{\sum_{i \in A} n_i^2}{N_A} \right) > \mu_e + \sigma^2 \frac{\sum_{i \in A} n_i^2}{N} - \sigma^2 \frac{\sum_{i \in A} n_i^2}{N_A}$$

Multiplying on the left side:

$$\frac{N_A}{N} \cdot \sigma^2 \cdot \frac{\sum_{i \in C \setminus A} n_i^2}{N} + \frac{N_A}{N} \cdot \left( \mu_e + \sigma^2 \cdot \frac{\sum_{i \in A} n_i^2}{N} - \sigma^2 \frac{\sum_{i \in A} n_i^2}{N_A} \right) > \mu_e + \sigma^2 \frac{\sum_{i \in A} n_i^2}{N} - \sigma^2 \frac{\sum_{i \in A} n_i^2}{N_A}$$

Collecting terms:

$$\frac{N_A}{N} \cdot \sigma^2 \cdot \frac{\sum_{i \in C \setminus A} n_i^2}{N} + \frac{N_A}{N} \cdot \left( \mu_e + \sigma^2 \cdot \left( \sum_{i \in A} n_i^2 \right) \cdot \left( \frac{1}{N} - \frac{1}{N_A} \right) \right) > \mu_e + \sigma^2 \cdot \left( \sum_{i \in A} n_i^2 \right) \cdot \left( \frac{1}{N} - \frac{1}{N_A} \right)$$

Changing the sign:

$$\frac{N_A}{N} \cdot \sigma^2 \cdot \frac{\sum_{i \in C \setminus A} n_i^2}{N} + \frac{N_A}{N} \cdot \left( \mu_e - \sigma^2 \cdot \left( \sum_{i \in A} n_i^2 \right) \cdot \left( \frac{1}{N_A} - \frac{1}{N} \right) \right) > \mu_e - \sigma^2 \cdot \left( \sum_{i \in A} n_i^2 \right) \cdot \left( \frac{1}{N_A} - \frac{1}{N} \right)$$

Bringing across terms to the righthand side:

$$\frac{N_A}{N} \cdot \sigma^2 \cdot \frac{\sum_{i \in C \setminus A} n_i^2}{N} > \left( 1 - \frac{N_A}{N} \right) \cdot \left( \mu_e - \sigma^2 \cdot \left( \sum_{i \in A} n_i^2 \right) \cdot \left( \frac{1}{N_A} - \frac{1}{N} \right) \right)$$

Bringing all coefficients of $\sigma^2$ to the lefthand side:

$$\frac{N_A}{N} \cdot \sigma^2 \cdot \frac{\sum_{i \in C \setminus A} n_i^2}{N} + \left( 1 - \frac{N_A}{N} \right) \cdot \sigma^2 \cdot \left( \sum_{i \in A} n_i^2 \right) \cdot \left( \frac{1}{N_A} - \frac{1}{N} \right) > \left( 1 - \frac{N_A}{N} \right) \cdot \mu_e$$

Rewriting:

$$\frac{N_A}{N} \cdot \sigma^2 \cdot \left( \sum_{i \in C \setminus A} n_i^2 \right) + (N - N_A) \cdot \sigma^2 \cdot \left( \sum_{i \in A} n_i^2 \right) \cdot \left( \frac{1}{N_A} - \frac{1}{N} \right) > (N - N_A) \cdot \mu_e$$

We strategically rewrite the righthand side:

$$\frac{N_A}{N} \cdot \sigma^2 \cdot \left( \sum_{i \in C \setminus A} n_i^2 \right) + (N - N_A) \cdot \sigma^2 \cdot \left( \sum_{i \in A} n_i^2 \right) \cdot \left( \frac{1}{N_A} - \frac{1}{N} \right) > (N - N_A) \cdot \mu_e \cdot \frac{N_A}{N} + (N - N_A) \cdot \left( 1 - \frac{N_A}{N} \right) \cdot \mu_e$$

$$\frac{N_A}{N} \cdot \sigma^2 \cdot \left( \sum_{i \in C \setminus A} n_i^2 \right) + (N - N_A) \cdot \sigma^2 \cdot \left( \sum_{i \in A} n_i^2 \right) \cdot \left( \frac{1}{N_A} - \frac{1}{N} \right) > (N - N_A) \cdot \mu_e \cdot \frac{N_A}{N} + (N - N_A) \cdot N_A \cdot \left( \frac{1}{N_A} - \frac{1}{N} \right) \cdot \mu_e$$

We pull all of the terms over to the lefthand side:

$$\frac{N_A}{N} \cdot \left( \sum_{i \in C \setminus A} n_i \cdot \left( \sigma^2 \cdot n_i - \mu_e \right) \right) + (N - N_A) \cdot \left( \frac{1}{N_A} - \frac{1}{N} \right) \cdot \left( \sum_{i \in A} n_i \cdot \left( n_i \cdot \sigma^2 - \mu_e \right) \right) > 0$$

Finally, we will show that the above inequality cannot hold. By assumption, $n_i \leq \frac{\mu_e}{\sigma^2}$ for all $i \in C$. This means that $\sigma^2 \cdot n_i - \mu_e$ is negative for all $i \in C$. Because every other term on the lefthand side is positive (note that $\frac{1}{N_A} > \frac{1}{N}$), we know that the lefthand term is negative. However, the inequality is requiring that the term is positive. By this contradiction, we know that the initial assumption must have been wrong: so long as $n_i \leq \frac{\mu_e}{\sigma^2}$, there cannot be any set $A$ such that each player strictly prefers $A$ to $C$, so the grand coalition $C$ is core stable. □

## C  Price of Anarchy

**Lemma 11.** *For a set of players with $n_i \geq \frac{\mu_e}{\sigma^2} \forall i$, any arrangement that is core stable or individually stable is also optimal.*

| Type | Condition | Upper bound on $err_i(\Pi_M)$ | Lower bound on $err_i(\Pi_{opt})$ |
|---|---|---|---|
| $T_0$ | $n_i \geq \frac{\mu_e + \sigma^2}{2 \cdot \sigma^2}$ | $\frac{\mu_e}{n_i}$, by Lemma 12. | $\frac{1}{2}\frac{\mu_e}{n_i}$, by Lemma 13 |
| $T_1$ | $\frac{\mu_e}{9 \cdot \sigma^2} \leq n_i \leq \frac{\mu_e + \sigma^2}{2\sigma^2}$ | | |
| $T_2$ | $n_i < \frac{\mu_e}{9 \cdot \sigma^2}$ and is federating with other players of total mass at least $\frac{\mu_e}{3\sigma^2}$ in $\Pi_M$. | $7.25 \cdot \sigma^2$, by Lemma 14 | $\sigma^2$, by Lemma 13 |
| $T_3$ | $n_i < \frac{\mu_e}{9 \cdot \sigma^2}$ and is NOT federating with other players of total mass at least $\frac{\mu_e}{3\sigma^2}$ in $\Pi_M$. | Unbounded, but Lemma 15 gives a stability result. | |

Table 2: Summary of relevant bounds for proof of Theorem 2.

*Proof.* By Lemma 5.3 in Donahue and Kleinberg [2021], when all players have $\geq \frac{\mu_e}{\sigma^2}$ samples, each player with size $> \frac{\mu_e}{\sigma^2}$ minimizes its error by doing local learning. By the same lemma, each player of size exactly equal to $\frac{\mu_e}{\sigma^2}$ minimize their error in any arrangement with other players also of size $\frac{\mu_e}{\sigma^2}$. Taken together, this implies that the only stable arrangements are ones where all players of size $> \frac{\mu_e}{\sigma^2}$ are doing local learning and all players of size equal $\frac{\mu_e}{\sigma^2}$ are arranged in any grouping. Because all of these have equal error to the minimal error, the Price of Anarchy is equal to 1. □

**Theorem 2** (Price of Anarchy). *Denote $\Pi_M$ to be a maximum-cost individually stable (IS) partition and $\Pi_{opt}$ to be an optimal (lowest-cost) partition. Then,*

$$PoA = \frac{f_w(\Pi_M)}{f_w(\Pi_{opt})} \leq 9$$

*Proof.* This theorem is the result of multiple lemmas, each of which handle players of different sizes in different situations. Theorem 2 summarizes these contributions. Specifically, it divides players into four different types ($T_0, T_1, T_2, T_3$) based on their size and the group they are federating with in $\Pi_M$. These results are summarized in Table 2 and described below.

First, we note that by Lemma 12 the highest error any player can experience in $\Pi_M$ is $\frac{\mu_e}{n_i}$, so the cost due to a particular player in $\Pi_M$ is upper bounded by $\mu_e$.

- Say that player $i \in T_0$ if $n_i \geq \frac{\mu_e + \sigma^2}{2\sigma^2}$. Lemma 13 shows that if $n_i \geq \frac{\mu_e + \sigma^2}{2\sigma^2}$, then $err_i(\Pi_{opt}) \geq \frac{1}{2}\frac{\mu_e}{n_i}$, so player $i$'s contribution to the weighted cost is $\geq \frac{1}{2} \cdot \mu_e$.

- Say that player $i \in T_1$ if $\frac{\mu_e}{9 \cdot \sigma^2} \leq n_i \leq \frac{\mu_e + \sigma^2}{2\sigma^2}$. Lemma 13 shows that $err_i(\Pi_{opt}) \geq \sigma^2$ for $n_i \leq \frac{\mu_e + \sigma^2}{2\sigma^2}$, so player $i$'s contribution to the weighted cost is $\geq \sigma^2 \cdot n_i$.

- Say that player $i \in T_2$ if $n_i < \frac{\mu_e}{9 \cdot \sigma^2}$ and if, in $\Pi_M$, it is federating with other players of total mass at least $\frac{\mu_e}{3\sigma^2}$. Then, by Lemma 14 $err_i(\Pi_M) \leq 7.25\sigma^2 \leq 7.5 \cdot \sigma^2$. Lemma 13 applies again and shows that $err_i(\Pi_{opt}) \geq \sigma^2$ for $n_i \leq \frac{\mu_e + \sigma^2}{2\sigma^2}$, so player $i$'s contribution to the weighted cost is $\geq \sigma^2 \cdot n_i$.

- Say $i \in T_3$ if $n_i \leq \frac{\mu_e}{9 \cdot \sigma^2}$ and if in $\Pi_M$ it is *not* federating with other players of total mass at least $\frac{\mu_e}{3 \cdot \sigma^2}$. Then, by Lemma 15 there is at most one group of such description in $\Pi_M$ (or any IS arrangement) - call it $A$. What is this group's total contribution to the cost?

$$\mu_e + \sigma^2 \cdot N_A - \sigma^2 \frac{\sum_{i \in A} n_i^2}{N_A} \leq \mu_e + \sigma^2 \cdot N_A - \sigma^2 \frac{N_A}{N_A} \leq^* \left(1 + \frac{1}{3} + \frac{1}{9}\right)\mu_e - \sigma^2 < 1.5\mu_e$$

where in the step marked with $*$ we have upper bounded $N_T$ by the knowledge that it contains a player of size $\leq \frac{\mu_e}{9\sigma^2}$ is federating with partners of total size no more than $\frac{\mu_e}{3\sigma^2}$. Note that $N_T$ is the mass of the entire group containing $T_3$ players, and so may double-count the contributions of some players not in $T_3$.

Next, we bring these terms together to bound the overall result. Note that $f_w(\Pi)$ is a weighted cost that is obtained by multiplying player $j$'s error by its number of samples $n_j$.

$$PoA = \frac{f_w(\Pi_M)}{f_w(\Pi_{opt})} \leq \frac{|T_0| \cdot \mu_e + |T_1| \cdot \mu_e + \sum_{i \in T_2} 7.5 \cdot \sigma^2 \cdot n_i + 1.5\mu_e}{|T_0| \cdot \frac{\mu_e}{2} + \sum_{i \in T_1} \sigma^2 \cdot n_i + \sum_{i \in T_2} \sigma^2 \cdot n_i + \sum_{i \in T_3} \sigma^2 \cdot n_i}$$

First, we note that if there do not exist any players in $T_3$, then we can write the bound as:

$$\frac{|T_0| \cdot \mu_e + |T_1| \cdot \mu_e + \sum_{i \in T_2} 7.5 \cdot \sigma^2 \cdot n_i}{|T_0| \cdot \frac{\mu_e}{2} + |T_1| \cdot \frac{\mu_e}{9} + \sum_{i \in T_2} \sigma^2 \cdot n_i} \leq 9$$

Suppose that $|T_3| \geq 1$. Then, the main goal is to absorb the additive $1.5 \cdot \mu_e$ term.

First, we consider the case where we have some player $n_j \geq \frac{\mu_e}{3\sigma^2}$, which we will show implies a PoA bound of 9. Any player of size $\geq \frac{\mu_e}{3 \cdot \sigma^2}$ must be in $T_0$ or $T_1$. First, we will assume that $j \in T_0$, so $|T_0| \geq 1$, meaning:

$$4.5 \cdot |T_0| \cdot \mu_e \geq |T_0| \cdot \mu_e + 1.5 \cdot \mu_e$$

This means the bound can be upper bounded by:

$$PoA \leq \frac{4.5|T_0| \cdot \mu_e + |T_1| \cdot \mu_e + \sum_{i \in T_2} 7.5 \cdot \sigma^2 \cdot n_i}{|T_0| \cdot \frac{\mu_e}{2} + |T_1| \cdot \frac{\mu_e}{9\sigma^2} + \sum_{i \in T_2} \sigma^2 \cdot n_i} \leq 9$$

Next, we consider the case where $j \in T_1$ and $|T_0| = 0$. Then, the upper bound becomes:

$$\begin{aligned}
PoA &< \frac{(|T_1| - 1) \cdot \mu_e + \mu_e + 7.5\sigma^2 \cdot \sum_{i \in T_2} n_i + 1.5\mu_e}{\sum_{i \neq j, i \in T_1} \sigma^2 \cdot n_i + \sigma^2 \cdot n_j + \sigma^2 \cdot \sum_{i \in T_2} n_i} \\
&< \frac{(|T_1| - 1) \cdot \mu_e + \mu_e + 7.5\sigma^2 \cdot \sum_{i \in T_2} n_i + 1.5\mu_e}{(|T_1| - 1) \cdot \frac{\mu_e}{9} + \frac{\mu_e}{3} + \sigma^2 \cdot \sum_{i \in T_2} n_i} \\
&< \frac{(|T_1| - 1) \cdot \mu_e + 2.5\mu_e + 9\sigma^2 \cdot \sum_{i \in T_2} n_i}{(|T_1| - 1) \cdot \frac{\mu_e}{9} + \frac{\mu_e}{3} + \sigma^2 \cdot \sum_{i \in T_2} n_i} \\
&< \frac{(|T_1| - 1) \cdot \mu_e + 3\mu_e + 9\sigma^2 \cdot \sum_{i \in T_2} n_i}{(|T_1| - 1) \cdot \frac{\mu_e}{9} + \frac{\mu_e}{3} + \sigma^2 \cdot \sum_{i \in T_2} n_i} \\
&= 9
\end{aligned}$$

Finally, we consider the case where all players have size $\leq \frac{\mu_e}{3\sigma^2}$. By Lemma 15, if there exist any players in $T_3$, then the entire arrangement is only stable if $\Pi_M = \pi_g = \Pi_{opt}$, giving a PoA of 1.

These proofs taken together show that the overall PoA is upper bounded by 9. $\square$

**Lemma 13.** *Consider a player $n_j$ and any set of players $C$. Then, we can lower bound the error player $j$ recieves by federating with $C$:*

$$err_j(C \cup \{n_j\}) \geq \begin{cases} \frac{1}{2} \cdot \frac{\mu_e}{n_j} & n_j \geq \frac{\mu_e + \sigma^2}{2\sigma^2} \\ \sigma^2 & \text{otherwise} \end{cases}$$

*Proof.* Player $j$'s error when federating with the coalition $C$ is:

$$err_j(C \cup \{n_j\}) = \frac{\mu_e}{N_C + n_j} + \sigma^2 \frac{\sum_{i \in C} n_i^2 + N_C^2}{(N_C + n_j)^2}$$

Given a fixed $N_C$, $\sum_{i \in C} n_i^2$ is minimized when all of the players besides $j$ have size $n_i = \frac{N_C}{|C|}$, which means that $n_i^2 = \frac{N_C^2}{|C|^2}$. The error is thus lower bounded by:

$$err_j(C \cup \{n_j\}) \geq \frac{\mu_e}{N_C + n_j} + \sigma^2 \frac{\frac{N_C^2}{|C|} + N_C^2}{(N_C + n_j)^2}$$

This decreases with $|C|$, so we set $|C| = N_C$ to further lower bound the error:

$$\geq \frac{\mu_e}{N_C + n_j} + \sigma^2 \frac{N_C + N_C^2}{(N_C + n_j)^2}$$

Note that the "units" of this term might seem strange: the numerator of the $\sigma^2$ component involves a $N_C$ and $N_C^2$. This is because we assumed that $\sum_{i \in C} n_i^2 \geq N_C$, which is correct in magnitude but which involves different units.

Next, we will lower bound this term by analyzing how it changes with $N_C$. First, we take the derivative with respect to $N_C$:

$$\frac{n_j \cdot (\sigma^2 - \mu_e + 2\sigma^2 \cdot N_C) - N_C \cdot (\mu_e + \sigma^2)}{(N_C + n_j)^3}$$

**Case 1: Derivative always negative**
In some situations, this derivative is always negative (the player $j$ always prefers $N_C$ as large as possible). When does this occur?

$$n_j \cdot (\sigma^2 - \mu_e + 2N_C \cdot \sigma^2) < (\mu_e + \sigma^2) \cdot N_C \quad \forall N_C$$

As $N_C \to \infty$, the $\sigma^2 - \mu_e$ additive term on the lefthand side becomes irrelevant, so what we require is

$$2\sigma^2 \cdot n_j \cdot N_C \leq (\mu_e + \sigma^2) \cdot N_C$$

$$n_j \leq \frac{\mu_e + \sigma^2}{2\sigma^2}$$

For players satisfying this premise, we can lower bound their error by sending $N_C \to \infty$ in the original error equation.

$$\lim_{N_C \to \infty} \left[ \frac{\mu_e}{N_C + n_j} + \sigma^2 \frac{N_C + N_C^2}{(N_C + n_j)^2} \right] = \sigma^2$$

This implies that the player's error goes to $\sigma^2$ (from above), so is lower bounded by $\sigma^2$.
**Case 2: Derivative sometimes negative, sometimes positive**

Next, we'll consider the case where $n_j > \frac{\mu_e + \sigma^2}{2\sigma^2}$. The second derivative of the player's error with respect to $N_C$ is:

$$2 \cdot \sigma^2 \cdot n_j - \mu_e - \sigma^2$$

which is greater than or equal to 0 in this case. In order to lower bound the overall error, we must bound the error when $N_C = 0$ (at its minimum value) and when the derivative with respect to $N_C$ is 0 (local minimum). Note that when $N_C = 0$, player $j$'s error is $\frac{\mu_e}{n_j}$, which is $> \frac{1}{2} \cdot \frac{\mu_e}{n_j}$, satisfying the premise. Next, we will consider the case where the derivative is equal to 0: In this case, the slope isn't always negative, so there must be some $N_C$ such that the slope is equal to 0. This occurs when:

$$n_j \cdot (\sigma^2 - \mu_e) + N_C \cdot (2n_j \cdot \sigma^2 - \mu_e - \sigma^2) = 0$$

$$N_C = \frac{n_j \cdot (\mu_e - \sigma^2)}{2n_j \cdot \sigma^2 - \mu_e - \sigma^2}$$

Substituting in for this value of $N_C$ into player $j$'s error gives:

$$\frac{-\mu_e^2 - 2\mu_e \cdot \sigma^2 + 4n_j \cdot \mu_e \cdot \sigma^2 - (\sigma^2)^2}{-4n_j \cdot \sigma^2 + 4n_j^2 \cdot \sigma^2} = \frac{\mu_e}{n_j} \cdot \frac{-\mu_e - 2\sigma^2 + 4n_j \cdot \sigma^2 - \sigma^2 \cdot \frac{\sigma^2}{\mu_e}}{-4 \cdot \sigma^2 + 4n_j \cdot \sigma^2}$$

In order to prove that this whole term is lower bounded by $\frac{1}{2} \frac{\mu_e}{n_j}$, we will show that the coefficient on $\frac{\mu_e}{n_j}$ is lower bounded by $\frac{1}{2}$. Because $n_j \geq 1$, we know that the denominator is positive:

$$\frac{-\mu_e - 2\sigma^2 + 4n_j \cdot \sigma^2 - \sigma^2 \cdot \frac{\sigma^2}{\mu_e}}{-4 \cdot \sigma^2 + 4n_j \cdot \sigma^2} \geq \frac{1}{2}$$

$$-2\mu_e - 4\sigma^2 + 8n_j \cdot \sigma^2 - \frac{\sigma^4}{\mu_e} \geq -4\sigma^2 + 4n_j \cdot \sigma^2$$

$$-2\mu_e + 4n_j \cdot \sigma^2 - \frac{\sigma^4}{\mu_e} \geq 0$$

$$n_j \geq \frac{\mu_e}{2\sigma^2} + \frac{\sigma^2}{4\mu_e}$$

This is satisfied if the lower bound is smaller than or equal to $\frac{\mu_e + \sigma^2}{2\sigma^2}$. We can show this by noting that $\mu_e \geq \sigma^2$ for any avenue of interest (otherwise, $\frac{\mu_e}{\sigma^2} < 1$ and by Lemma 11 the only stable arrangement is to have all players doing local learning). This means that:

$$\frac{\mu_e}{2\sigma^2} + \frac{\sigma^2}{4\mu_e} \leq \frac{\mu_e}{2\sigma^2} + \frac{1}{4} = \frac{\mu_e + \frac{1}{2}\sigma^2}{2\sigma^2} < \frac{\mu_e + \sigma^2}{2\sigma^2}$$

as desired. This shows that:

$$err_j(C \cup \{n_j\}) \geq \frac{1}{2}\frac{\mu_e}{n_j}$$

$\square$

**Lemma 14.** *Consider a player $j$ federating with a coalition $C$. If the total number of samples $N_C$ is at least $\frac{\mu_e}{3\sigma^2}$, then $err_j(C \cup \{n_j\}) \leq 7.25 \cdot \sigma^2$.*

*Proof.* The error a player $n_j$ experiences is given by:

$$err_j(C \cup \{n_j\}) = \frac{\mu_e}{n_j + N_C} + \sigma^2 \frac{\sum_{i \in l} n_i^2 + N_C^2}{(N_C + n_j)^2}$$

Given a fixed total sum $N_C$, the $\sum_{i \in l} n_i^2$ term is maximized when all of the mass is on a single partner. So the overall cost can be upper bounded by:

$$< \frac{\mu_e}{N_C + n_j} + \sigma^2 \frac{2N_C^2}{(N_C + n_j)^2}$$

Taking the derivative with respect to $N_C$ gives:

$$-\frac{\mu_e}{(N_C + n_j)^2} + \sigma^2 \frac{4N_C \cdot (N_C + n_j)^2 - 4N_C^2 \cdot (N_C + n_j)}{(N_C + n_j)^4} = -\frac{\mu_e}{(N_C + n_j)^2} + \sigma^2 \frac{4N_C \cdot n_j}{(N_C + n_j)^3}$$

$$= \frac{-\mu_e \cdot (N_C + n_j) + 4\sigma^2 N_C \cdot n_j}{(N_C + n_j)^3}$$

Next, we will upper bound player $j$'s error based on the sign of the derivative with respect to $N_C$.

**Case 1: Derivative with respect to $N_C$ always positive**:
This occurs when the numerator is positive for all $N_C \geq 0$, or

$$-\mu_e \cdot (N_C + n_j) + 4\sigma^2 N_C \cdot n_j > 0$$

$$N_C \cdot (4\sigma^2 \cdot n_j - \mu_e) > \mu_e \cdot n_j$$

To begin with, we must have that $4\sigma^2 \cdot n_j > \mu_e$ or else the lefthand side is negative, so $n_j > \frac{\mu_e}{4\sigma^2}$. Given that, the error is largest when $N_C$ is set to its largest value of $\frac{\mu_e}{3\sigma^2}$.

$$\frac{\mu_e}{3 \cdot \sigma^2} \cdot (4\sigma^2 n_j - \mu_e) > \mu_e \cdot n_j$$

$$4\sigma^2 n_j - \mu_e > 3\sigma^2 \cdot n_j$$

$$n_j > \frac{\mu_e}{\sigma^2}$$

If this is the case, what is the maximum amount of error that $n_j$ receives? The error is in the form:

$$\frac{\mu_e}{N_C + n_j} + \sigma^2 \frac{2N_C^2}{(N_C + n_j)^2}$$

We know that this is maximized when $N_C \to \infty$. In this case, $\mu_e$ term goes to 0. The $\sigma^2$ term (by L'Hôpital's rule) goes to:

$$\sigma^2 \frac{4N_C}{2(N_C + n_j)} \to 2\sigma^2$$

**Case 2: Derivative with respect to $N_C$ is always negative**
Next, we'll consider the inverse case where the derivative is always negative. This occurs when:

$$N_C \cdot (4\sigma^2 \cdot n_j - \mu_e) < \mu_e \cdot n_j \quad \forall N_C$$

This has to be true for all $N_C$, which implies that the $4\sigma^2 \cdot n_j - \mu_e$ term is negative, or $n_j \leq \frac{\mu_e}{4\sigma^2}$. If this is the case, the maximal error is achieved when the $N_C$ term is smallest ($\frac{\mu_e}{3 \cdot \sigma^2}$). Plugging into the error form gives us:

$$
\frac{\mu_e}{\frac{\mu_e}{3 \cdot \sigma^2} + n_j} + \sigma^2 \frac{2 \cdot \frac{\mu_e^2}{9 \cdot \sigma^4}}{\left(\frac{\mu_e}{3 \cdot \sigma^2} + n_j\right)^2} = \frac{\mu_e \cdot \left(\frac{\mu_e}{3 \cdot \sigma^2} + n_j\right) + \frac{2\mu_e^2}{9 \cdot \sigma^2}}{\left(\frac{\mu_e}{3 \cdot \sigma^2} + n_j\right)^2}
$$

$$
= \frac{\mu_e \cdot \left(\frac{\mu_e}{3 \cdot \sigma^2} + n_j\right) + \frac{2\mu_e^2}{9 \cdot \sigma^2}}{\frac{1}{9\sigma^4} \cdot \left(\mu_e + 3\sigma^2 \cdot n_j\right)^2}
$$

$$
< \frac{9\sigma^4 \mu_e \cdot \left(\frac{\mu_e}{3 \cdot \sigma^2} + n_j\right) + 2 \cdot \mu_e^2 \cdot \sigma^2}{\mu_e^2}
$$

$$
= 3\sigma^2 + 9\sigma^2 \cdot \frac{\sigma^2}{\mu_e} \cdot n_j + 2\sigma^2
$$

$$
< 5\sigma^2 + 9\sigma^2 \cdot \frac{\sigma^2}{\mu_e} \cdot \frac{\mu_e}{4\sigma^2}
$$

$$
= 7.25
$$

where in the last step we have used that $n_j \leq \frac{\mu_e}{4\sigma^2}$.

**Case 3: when the derivative with respect to $N_C$ is sometimes positive and sometimes negative**

Using the values above, we know this occurs when $\frac{\mu_e}{4\sigma^2} \leq n_j \leq \frac{\mu_e}{\sigma^2}$. First, we'll confirm that the error first decreases and then increases with $N_C$. The derivative is:

$$
N_C \cdot (4\sigma^2 \cdot n_j - \mu_e) - \mu_e \cdot n_j
$$

Here, we are assuming that the coefficient on $N_C$ is either 0 or positive, so the second derivative with respect to $N_C$ is positive. Given that the derivative is negative at some point, it must be negative for small $N_C$. We know from Case 1 that as $N_C \to \infty$, the error goes to $2\sigma^2$, so in order to bound the entire space, we only need to bound the error at the smallest value of $N_C$, which is $\frac{\mu_e}{3 \cdot \sigma^2}$. The first few steps are identical to Case 2:

$$
\frac{\mu_e}{\frac{\mu_e}{3 \cdot \sigma^2} + n_j} + \sigma^2 \frac{2 \cdot \frac{\mu_e^2}{9 \cdot \sigma^4}}{\left(\frac{\mu_e}{3 \cdot \sigma^2} + n_j\right)^2} = \frac{\mu_e \cdot \left(\frac{\mu_e}{3 \cdot \sigma^2} + n_j\right) + \frac{2\mu_e^2}{9 \cdot \sigma^2}}{\left(\frac{\mu_e}{3 \cdot \sigma^2} + n_j\right)^2} = \frac{\mu_e \cdot \left(\frac{\mu_e}{3 \cdot \sigma^2} + n_j\right) + \frac{2\mu_e^2}{9 \cdot \sigma^2}}{\frac{1}{9\sigma^4} \cdot \left(\mu_e + 3\sigma^2 \cdot n_j\right)^2}
$$

In the next step, though, we use that $\frac{\mu_e}{4\sigma^2} \leq n_j \leq \frac{\mu_e}{\sigma^2}$.

$$
< \frac{9\sigma^4 \mu_e \cdot \left(\frac{\mu_e}{3 \cdot \sigma^2} + n_j\right) + 2 \cdot \mu_e^2 \cdot \sigma^2}{(\mu_e + \frac{3}{4}\mu_e)^2}
$$

$$
= \frac{3\sigma^2 + 9\sigma^2 \cdot \frac{\sigma^2}{\mu_e} \cdot n_j + 2\sigma^2}{\frac{49}{16}}
$$

$$
< \frac{16}{49} \cdot \left(5\sigma^2 + 9\sigma^2 \cdot \frac{\sigma^2}{\mu_e} \cdot \frac{\mu_e}{\sigma^2}\right)
$$

$$
= \frac{16}{49} \cdot 14 \cdot \sigma^2
$$

$$
< 5\sigma^2
$$

Of the three cases, the highest bound is $7.25 \cdot \sigma^2$. $\qquad \square$

Lemma 15, below, relies on Lemmas 16, 17, and 18, which are stated and proved immediately after the proof of Lemma 15.

**Lemma 15.** *Consider an arrangement of players, all of size $\leq \frac{\mu_e}{3\sigma^2}$, where at least one player is in a federating cluster where the total mass of its partners is no more than $\frac{\mu_e}{3\sigma^2}$. Then, the only stable arrangement of these players is to have all of them federating together.*

*Proof.* By Lemma 16, we know that every player in every group welcomes the addition of any other player. Therefore, in order to prove that this arrangement isn't individually stable, we simply have to prove that a player would wish to move.

We will consider a cluster $A$ with elements $i \in T_3$ present. We know that there exists at least one element in $A$ s.t. the mass of its partners ($N_A - n_i$) is less than $\frac{\mu_e}{3\sigma^2}$. This implies also that $N - n_a < \frac{\mu_e}{3\sigma^2}$ for $n_a$ the largest element in $A$. We also know that $n_a < \frac{\mu_e}{3\sigma^2}$ because we know that there exists some other element in the cluster with $N_A - n_i < \frac{\mu_e}{3\sigma^2}$.

Next, let's suppose there exists some other cluster $B$, such that all elements are $\leq \frac{\mu_e}{3\sigma^2}$ in size. We will consider some $n_b$ largest player in $B$. There are four possible cases:

1. $n_a \geq n_b, N_A - n_a \geq N_B - n_b$: Unstable by Lemma 17 (player $b$ wishes to move to $A$).

2. (Symmetric to above) $n_a \leq n_b, N_A - n_a \leq N_B - n_b$: Unstable by Lemma 17 (player $a$ wishes to move to $B$).

3. $n_a > n_b, N_A - n_a < N_B - n_b$. Note that in this case, we know that $N_A - n_a \leq \frac{\mu_e}{3\sigma^2}$, so we satisfy the conditions of Lemma 18, and thus player $a$ would prefer to join $B$.

4. $n_a < n_b, N_A - n_a > N_B - n_b$. In this case, we know that $\frac{\mu_e}{3\sigma^2} > N_A - n_a > N_B - n_b$, so we again satisfy the conditions of Lemma 18, and thus player $b$ would prefer to join $A$.

$\square$

**Lemma 16.** *A group of players where each has size $n_i \leq \frac{\mu_e}{3\sigma^2}$ always welcomes the addition of another player of size $n_k \leq \frac{\mu_e}{3\sigma^2}$.*

*Proof.* For this section, we will rewrite the form of the error that a player experiences while federating with a coalition $C$. Specifically, we will write the error in the form below, where $a_i$ refers to the number of players with number of samples $n_i$.

$$\frac{\mu_e}{\sum_{i=1}^M a_i \cdot n_i} + \sigma^2 \frac{\sum_{i \neq j} a_i \cdot n_i^2 + (a_j - 1) \cdot n_j^2 + (\sum_{i \neq j} a_i \cdot n_i + (a_j - 1) \cdot n_j)^2}{(\sum_{i=1}^M a_i \cdot n_i)^2}$$

Setting $N = \sum_{i=1}^M a_i \cdot n_i$ gives:

$$\frac{\mu_e}{N} + \sigma^2 \cdot \frac{\sum_{i \neq j} a_i \cdot n_i^2 + (a_j - 1) \cdot n_j^2 + (N - n_j)^2}{N^2}$$

In order to prove that any player $j$ welcomes the addition of any other player $k$, we will show that the derivative with respect to $a_k$ is always negative. This means that player $j$ always sees its error decrease with the addition of another player of size $n_k$. As we take the derivative, the coefficient on the $\mu_e$ term in the error value becomes:

$$-\frac{\mu_e \cdot n_k}{N^2} = -\frac{\mu_e \cdot n_k \cdot N^2}{N^4}$$

The derivative of the coefficient on the $\sigma^2$ term becomes:

$$\frac{\sigma^2}{N^4} \cdot \left( (n_k^2 + 2(N - n_j) \cdot n_k) \cdot N^2 - \left( \sum_{i \neq j} a_i \cdot n_i^2 + (a_j - 1) \cdot n_j^2 + (N - n_j)^2 \right) \cdot 2 \cdot N \cdot n_k \right)$$

So, the overall derivative is negative if:

$$\mu_e \cdot n_k \cdot N^2 > \sigma^2 \cdot \left( (n_k^2 + 2(N - n_j) \cdot n_k) \cdot N^2 - \left( \sum_{i \neq j} a_i \cdot n_i^2 + (a_j - 1) \cdot n_j^2 + (N - n_j)^2 \right) \cdot 2 \cdot N \cdot n_k \right)$$

We pull out and cancel common terms:

$$\mu_e \cdot n_k \cdot N^2 > \sigma^2 \cdot n_k \cdot N \cdot \left( (n_k + 2N - 2n_j) \cdot N - 2 \left( \sum_{i \neq j} a_i \cdot n_i^2 + (a_j - 1) \cdot n_j^2 + (N - n_j)^2 \right) \right)$$

$$\mu_e \cdot N > \sigma^2 \cdot \left( (n_k + 2(N - n_j)) \cdot (N - n_j + n_j) - 2 \left( \sum_{i \neq j} a_i \cdot n_i^2 + (a_j - 1) \cdot n_j^2 + (N - n_j)^2 \right) \right)$$

Strategically expanding:

$$\mu_e \cdot N > \sigma^2 \cdot \left( n_k \cdot N + 2(N - n_j)^2 + 2n_j \cdot N - 2n_j^2 - 2 \left( \sum_{i \neq j} a_i \cdot n_i^2 + (a_j - 1) \cdot n_j^2 \right) - 2(N - n_j)^2 \right)$$

Collecting:

$$\mu_e \cdot N > \sigma^2 \cdot \left( N \cdot (n_k + 2n_j) - 2 \sum_{i=1}^{M} a_i \cdot n_i^2 \right)$$

Substituting in for $N$:

$$\mu_e \cdot \sum_{i=1}^{M} a_i \cdot n_i > \sigma^2 \cdot \left( \sum_{i=1}^{M} a_i \cdot n_i \cdot (n_k + 2n_j) - 2 \sum_{i=1}^{M} a_i \cdot n_i^2 \right)$$

$$0 > \sum_{i=1}^{M} a_i \cdot n_i \cdot (\sigma^2 \cdot n_k + 2\sigma^2 \cdot n_j - 2\sigma^2 \cdot n_i - \mu_e)$$

Our goal is to show that this is negative if $n_i \leq \frac{\mu_e}{3\sigma^2}$ for all $i$.

First, we look over the portion of the sum equal to the $k$ index. This term is equal to:

$$a_k \cdot n_k \cdot (2\sigma^2 \cdot n_j - \sigma^2 \cdot n_k - \mu_e)$$

which is negative, given our conditions. Next, we look at the $j$ term in the sum:

$$a_j \cdot n_j \cdot (\sigma^2 \cdot n_k - \mu_e)$$

which is also negative. The remaining portions of the sum can be written as:

$$(N - a_j \cdot n_j - a_k \cdot n_k) \cdot (\sigma^2 \cdot n_k + 2\sigma^2 \cdot n_j - \mu_e) - 2\sigma^2 \sum_{i \neq j,k} a_i \cdot n_i^2$$

which we would like to show is negative. We can maximize this term by holding $N$ constant and minimizing the negative portion by setting $n_i = 1$ for all other players besides $j, k$. This gives us an upper bound of:

$$\leq (N - a_j \cdot n_j - a_k \cdot n_k) \cdot (\sigma^2 \cdot n_k + 2\sigma^2 \cdot n_j - \mu_e) - 2\sigma^2(N - a_j \cdot n_j - a_k \cdot n_k)$$
$$= (N - a_j \cdot n_j - a_k \cdot n_k) \cdot (\sigma^2 \cdot n_k + 2\sigma^2 \cdot n_j - \mu_e - 2\sigma^2)$$

Given the condition that $n_k, n_j \leq \frac{\mu_e}{3\sigma^2}$, we know that the coefficient is no more than

$$3\sigma^2 \frac{\mu_e}{3\sigma^2} - \mu_e - 2\sigma^2 < 0$$

Taken together, this shows that the derivative of player $j$'s error with respect to $a_k$ is negative, which means that player $j$ always sees its error decrease with the addition of another player $k$. $\square$

**Lemma 17.** *Assume we have two groups of players, $A$ and $B$ with all players of size $\leq \frac{\mu_e}{3\sigma^2}$. Then, if either of the two conditions below are satisfied, the arrangement is not individually stable.*

1. *There exists $a \in A, b \in B$ such that $n_a = n_b$.*

2. *There exists $a \in A, b \in B$ such that $n_a > n_b$ and $N_A - n_a \geq N_B - n_b$. (Note that this could be defined symmetrically with respect to $B$).*

*Proof.* First, we will assume that player $a$ does not wish to move to $B$ (if this is not true, then we already know that the arrangement is not IS). This tells us that:

$$err_a(A) \leq err_a(B \cup \{n_a\})$$

Next, we will derive sufficient conditions for player $b$ to wish to move to $A$, or

$$err_b(A \cup \{n_b\}) < err_b(B)$$

We will use the shorthand of $N'_A = N_A - n_a$ and $N'_B = N_B - n_b$. From the form of each player's error as in Lemma 1, we can derive conditions for the difference in errors experienced by two players in the same coalition. Consider a coalition $C$ and two players $j, k \in C$, with $n_k \geq n_j$ Then,

$$
\begin{aligned}
err_j(C) - err_k(C) &= \sigma^2 \cdot \frac{\sum_{i \neq j} n_i^2 + (N_C - n_j)^2}{N_C^2} - \sigma^2 \cdot \frac{\sum_{i \neq k} n_i^2 + (N_C - n_k)^2}{N_C^2} \\
&= \sigma^2 \cdot \frac{n_k^2 - n_j^2 + (N_C - n_j)^2 - (N_C - n_k)^2}{N_C^2} \\
&= \sigma^2 \cdot \frac{n_k^2 - n_j^2 + (N_C^2 + n_j^2 - 2n_j \cdot N_C) - (N_C^2 + n_k^2 - 2n_k \cdot N_C)}{N_C^2} \\
&= \sigma^2 \cdot \frac{-2n_j \cdot N_C + 2n_k \cdot N_C}{N_C^2} \\
&= 2\sigma^2 \cdot \frac{N_C \cdot (n_k - n_j)}{N_C^2} \\
&= 2\sigma^2 \cdot \frac{n_k - n_j}{N_C}
\end{aligned}
$$

We can apply this derivation to obtain two equalities:

$$
err_b(A \cup b) = err_a(A \cup b) + 2\sigma^2 \frac{n_a - n_b}{N'_A + n_a + n_b}
$$

$$
err_a(B \cup a) = err_b(B \cup a) - 2\sigma^2 \frac{n_a - n_b}{N'_B + n_a + n_b}
$$

So, rewriting the first inequality tells us that:

$$
err_a(A) \leq err_b(B \cup a) - 2\sigma^2 \frac{n_a - n_b}{N'_B + n_a + n_b}
$$

Pulling over:

$$
err_a(A) + 2\sigma^2 \frac{n_a - n_b}{N'_B + n_a + n_b} \leq err_b(B \cup a)
$$

Note that because all of the players are of size $\leq \frac{\mu_e}{3 \cdot \sigma^2}$, we know by Lemma 16 that every player welcomes the addition of every other player, so

$$
err_b(B \cup a) < err_b(B)
$$

In order to complete the proof, we need to show that $err_b(A \cup \{n_b\})$ is less than $err_a(A) + 2\sigma^2 \frac{n_a - n_b}{N'_B + n_a + n_b}$. Again, because all of the players are of size $\leq \frac{\mu_e}{3 \cdot \sigma^2}$, we know from Lemma 16 that every player welcomes the addition of every other player, so

$$
err_a(A \cup \{n_b\}) + 2\sigma^2 \frac{n_a - n_b}{N'_B + n_a + n_b} < err_a(A) + 2\sigma^2 \frac{n_a - n_b}{N'_B + n_a + n_b}
$$

From our prior relation, we know that

$$
err_b(A \cup b) - 2\sigma^2 \frac{n_a - n_b}{N'_A + n_a + n_b} + 2\sigma^2 \frac{n_a - n_b}{N'_B + n_a + n_b} = err_a(A \cup \{n_b\}) + 2\sigma^2 \frac{n_a - n_b}{N'_B + n_a + n_b}
$$

Rewriting the term on the left tells us that what we want to show is:

$$
err_b(A \cup b) \leq err_b(A \cup b) + 2\sigma^2 \cdot (n_a - n_b) \cdot \left( \frac{1}{N'_B + n_a + n_b} - \frac{1}{N'_A + n_a + n_b} \right)
$$

Now, we can apply our case analysis. If $n_a = n_b$, then the added coefficient is 0, so the final inequality holds. The inequality also holds if the fractional coefficient is positive or 0, or

$$
\frac{1}{N'_B + n_a + n_b} \geq \frac{1}{N'_A + n_a + n_b}
$$

$$
N'_A + n_a + n_b \geq N'_B + n_a + n_b
$$

$$
N'_A \geq N'_B
$$

which is exactly the second criteria. $\qquad \square$

**Lemma 18.** *Assume we have two groups of players, A and B, with all players of size $\leq \frac{\mu_e}{3\sigma^2}$. Define $n_a, n_b$ to be the largest players in $A, B$ respectively. Assume that $n_a > n_b$ and $N_A - n_a < N_B - n_b$, with $N_A - n_a \leq \frac{\mu_e}{3\sigma^2}$. Then, player $a$ would prefer to join $B$.*

*Proof.* We will show that the preconditions imply that player $a$ would wish to move to group $B$, or else

$$err_a(B \cup \{n_a\}) < err_a(A)$$

Or, rewritten out,

$$\frac{\mu_e}{N_B + n_a} + \sigma^2 \frac{\sum_{i \in B} n_i^2 + N_B^2}{(N_B + n_a)^2} < \frac{\mu_e}{N_A} + \sigma^2 \frac{\sum_{i \in A, i \neq a} n_i^2 + (N_A - n_a)^2}{N_A^2}$$

We will upper and lower bound the costs on both sides by taking the worst and best case scenario for how the $B$ and $A$ players can be arranged, respectively. We have already showed that we can minimize the total arrangement of fixed total mass by dividing it into players of size exactly 1, so the player sizes equal to 1, or

$$\sum_{i \in A, i \neq a} n_i^2 \geq N_A - n_a$$

Conversely, let's try to upper bound the $B$ sum. Previously, we did this by grouping all of the mass into a single player. In this case, we can't do this - we've assumed that the $n_b$ term is the largest of them, so the most we can set them to be equal to is $n_b$ exactly. However, the same reasoning still holds: if we keep the total $N_B - n_b$ constant but rearrange them into groups of maximum size $b$, we only increase total cost. To see why, consider that we have $x, y$ with $x \geq y$, and some $x \leq b \leq x + y$. Then, we wish to show that:

$$x^2 + y^2 < b^2 + (x + y - b)^2$$

Expanding:

$$x^2 + y^2 < b^2 + (x + y - b)^2 = b^2 + b^2 + x^2 + y^2 - 2b \cdot x - 2b \cdot y + 2x \cdot y$$

Cancelling common terms means we want to show:

$$2b \cdot x + 2b \cdot y < 2b^2 + 2x \cdot y$$

$$x + y < b + \frac{x \cdot y}{b} < b + \frac{b \cdot y}{b} = b + y$$

which is satisfied.

This result tells us that this process (grouping them into players of exactly size $n_b$, plus at most one player of size $< n_b$) does maximize the total sum, subject to this constraint. We will again use the shorthand of $N_A' = N_A - n_a$ and $N_B' = N_B - n_b$. Excluding player $n_b$, the mass is $N_B'$, so the number of copies of $n_b$ that we can make is $\frac{N_B'}{n_b} := c + \epsilon$, for integer $c$ and $\epsilon \in [0, 1)$. If we know that $\epsilon = 0$ (which is always achievable), then we know that:

$$\sum_{i \in B, i \neq b} n_i^2 \leq c \cdot n_b^2 = \frac{N_B'}{n_b} \cdot n_b^2 = N_B' \cdot n_b$$

What if $\epsilon > 0$? Then,

$$\sum_{i \in B, i \neq b} n_i^2 \leq c \cdot n_b^2 + (\epsilon \cdot n_b)^2 < c \cdot n_b^2 + \epsilon \cdot n_b^2 = \frac{N_B'}{n_b} \cdot n_b^2 = N_B' \cdot n_b$$

So, in either way, the $N_B' \cdot n_b$ term is an upper bound. This means that the worst-case scenario for us to show that:

$$\frac{\mu_e}{N_B' + n_a + n_b} + \sigma^2 \frac{N_B' \cdot n_b + n_b^2 + (N_B' + n_b)^2}{(N_B' + n_a + n_b)^2} < \frac{\mu_e}{N_A' + n_a} + \sigma^2 \frac{N_A' + (N_A')^2}{(N_A' + n_a)^2}$$

We'll work by upper bounding the lefthand side. First, we'll replace the $N_B'$. First, we'll also look at the derivative with respect to $N_B'$, which gives:

$$\frac{n_a(-\mu_e + 3n_b \cdot \sigma^2 + 2N_b \cdot \sigma^2) - (n_b + N_B')(\mu_e + n_b\sigma^2)}{(n_a + n_b + N_B')^3}$$

The numerator can be rewritten as:

$$-\mu_e \cdot (n_a + n_b + N'_B) - \sigma^2 \cdot n_b \cdot (N'_B + n_b) + 3\sigma^2 \cdot n_a \cdot n_b + 2\sigma^2 \cdot n_a \cdot N'_B$$

We can show that this is negative because:

$$N'_B \cdot (-\mu_e + 2\sigma^2 \cdot n_a) < 0$$

since $n_a \leq \frac{\mu_e}{3\sigma^2}$. Similarly,

$$n_b \cdot (-\mu_e + 3\sigma^2 \cdot n_a) \leq 0$$

Because the derivative with respect to $N'_B$ is negative, we can over-bound it by setting it to its smallest value: $N'_A + 1$ (or $N'_A$, for simplicity). This means that we can upper bound the lefthand side by writing:

$$\frac{\mu_e}{N'_A + n_a + n_b} + \sigma^2 \frac{N'_A \cdot n_b + n_b^2 + (N'_A + n_b)^2}{(N'_A + n_a + n_b)^2} < \frac{\mu_e}{N'_A + n_a} + \sigma^2 \frac{N'_A + (N'_A)^2}{(N'_A + n_a)^2}$$

Next, we'll work on replacing the $n_b$ term on the lefthand side. We start out by taking the derivative of the lefthand side with respect to $n_b$. This gives us:

$$\frac{-\mu_e \cdot (n_a + N'_A) + \sigma^2 \cdot N'_A \cdot (N'_A + 3n_a) + n_b \cdot (-\mu_e + \sigma^2 \cdot (N'_A + 4n_a))}{(N'_A + n_a + n_b)^3}$$

We will inspect the sign of the derivative, which is given by the numerator. Specifically, we will show that the derivative is always negative or 0 at $n_b = 0$, and is either negative forever, or else is negative and then positive. This implies that the lefthand side of the overall equation is either always decreasing in $n_b$ (implying that we can upper bound it by setting $n_b = 0$) or else is decreasing and then increasing (in which case the upper bound is either at $n_b = 0$ or $n_b = n_a$).

First, we will prove our claim about the derivative. At $n_b = 0$, the derivative is:

$$-\mu_e \cdot (n_a + N'_A) + \sigma^2 \cdot N'_A \cdot (N'_A + 3n_a)$$

We want to show this is negative, or:

$$\sigma^2 \cdot N'_A \cdot (3n_a + N'_A) \leq \mu_e \cdot (n_a + N'_A)$$

Upper bounding the lefthand side:

$$3\sigma^2 \cdot N'_A \cdot (n_a + N'_A) \leq \mu_e \cdot (n_a + N'_A)$$

$$3\sigma^2 \cdot N'_A \leq \mu_e$$

which is satisfied by assumption. So, we know that the derivative starts out as 0 or negative. If the coefficient on $n_b$ (equal to $\sigma^2 \cdot (4n_a + N'_A) - \mu_e$) is negative, then the lefthand side of the overall equation is always decreasing as $n_b$ increases - so the upper bound at $n_b = 0$ suffices. Otherwise, the curve is decreasing, then increasing.

**Upper bound at $n_b = 0$**
This bound is fairly straightforward. What we want to show is:

$$\frac{\mu_e}{N'_A + n_a} + \sigma^2 \frac{N'^2_A}{(N'_A + n_a)^2} < \frac{\mu_e}{N'_A + n_a} + \sigma^2 \frac{N'_A + (N'_A)^2}{(N'_A + n_a)^2}$$

which is obviously true.

**Upper bound at $n_b = n_a$**
This bound is trickier. (Note that technically, the upper bound is at $n_a - 1$, but it is simpler to over-bound with $n_a$). What we'd like to show is:

$$\frac{\mu_e}{N'_A + 2n_a} + \sigma^2 \frac{N'_A \cdot n_a + n_a^2 + (N'_A + n_a)^2}{(N'_A + 2n_a)^2} \leq \frac{\mu_e}{N'_A + n_a} + \sigma^2 \frac{N'_A + N'^2_A}{(N'_A + n_a)^2}$$

We can write:

$$\mu_e \cdot \left( \frac{1}{N'_A + n_a} - \frac{1}{N'_A + 2n_a} \right) = \mu_e \cdot \frac{n_a}{(N'_A + n_a) \cdot (N'_A + 2n_a)}$$

Next, we move on to the $\sigma^2$ portion. Note that we can simplify the lefthand side, since:

$$N_A'^2 + n_a^2 + 2N_A' \cdot n_a + n_a^2 + N_A' \cdot n_a = N_A'^2 + 2n_a^2 + 3N_A' \cdot n_a = (N_A' + n_a) \cdot (N_A' + 2n_a)$$

So, the inequality we'd like to show becomes:

$$\sigma^2 \frac{N_A' + n_a}{N_A' + 2n_a} - \sigma^2 \frac{N_A' + N_A'^2}{(N_A' + n_a)^2} \leq \mu_e \cdot \frac{n_a}{(N_A' + n_a) \cdot (N_A' + 2n_a)}$$

Simplifying the lefthand side gives:

$$\sigma^2 \cdot \frac{(N_A' + n_a)^3 - (N_A' + N_A'^2) \cdot (N_A' + 2n_a)}{(N_A' + 2n_a) \cdot (N_A' + n_a)^2} \leq \mu_e \cdot \frac{n_a}{(N_A' + n_a) \cdot (N_A' + 2n_a)}$$

$$\sigma^2 \cdot \frac{(N_A' + n_a)^3 - (N_A' + N_A'^2) \cdot (N_A' + 2n_a)}{N_A' + n_a} \leq \mu_e \cdot n_a$$

We can make the lefthand side larger by making the negative part smaller - specifically, replacing the $(N_A' + 2n_a)$ with a $(N_A' + n_a)$. This gives us:

$$\sigma^2 \cdot \frac{(N_A' + n_a)^3 - (N_A' + N_A'^2) \cdot (N_A' + n_a)}{N_A' + n_a} \leq \mu_e \cdot n_a$$

$$\sigma^2 \cdot \left((N_A' + n_a)^2 - (N_A' + N_A'^2)\right) \leq \mu_e \cdot n_a$$

Expanding out the lefthand side gives us:

$$\sigma^2 \cdot (N_A'^2 + n_a^2 + 2n_a \cdot N_A' - N_A' - N_A'^2) \leq \mu_e \cdot n_a$$

$$\sigma^2 \cdot (n_a^2 + 2n_a \cdot N_A' - N_A') \leq \mu_e \cdot n_a$$

Again, we can make the lefthand side larger by dropping the negative portion:

$$\sigma^2 \cdot (n_a^2 + 2n_a \cdot N_A') \leq \mu_e \cdot n_a$$

$$\sigma^2 \cdot (n_a + 2N_A') \leq \mu_e$$

Which is satisfied because we require $N_A', n_a$ both $\leq \frac{\mu_e}{3\sigma^2}$. Note that, while this is a $\leq$, because we know that $n_b < n_a$, the overall inequality is strict. $\qquad\square$