# OpenReview forum: "Optimality and Stability in Federated Learning: A Game-theoretic Approach"
_NeurIPS.cc/2021/Conference — NeurIPS 2021 Poster_

### Official Review · Reviewer_cBBM · 2021-07-06

**Rating:** 7
**Confidence:** 4

**Summary:**

This paper studies the stable coalition in federated learning, where the agents can choose to join one coalition or leave with incentives to minimize their own errors. The paper focuses on the task of mean estimation, where the error has a closed-form solution. The main contributions are:
1. proposing an optimal algorithm to minimize the weighted sum of errors;
2. providing the upper bound on PoA.

**Limitations And Societal Impact:**

Yes

**Main Review:**

Originality: The task and the methods are new. The paper provides an optimal algorithm and an upper bound on PoA, which have not been studied before.

Quality: The submission is technically sound. All claims are well supported.

Clarity: The writing is clear. But some important notions should be addressed more formally, e.g., core stable and individually stable. It would be better if both notions could be defined mathematically. Besides, in line 261, the authors states "all members of $C$ weakly prefer...", where I don't understand what "weakly" means here.

Significance: The paper studies an interesting problem on the stable coalition in federated learning.
It looks like this paper only discusses the results of the mean estimation problem. In this specific task,  the error has a closed-form solution, which makes the analysis easier than that in a more general task. But it is still a good first step towards analyzing a more general problem. The results are important.  The paper provides an elegant algorithm achieving optimality and shows that PoA w.r.t. individually stability is upper bounded by 9.

Other comments:
1. The paper mentions that it focuses on mean estimation game in Line 163. If this submission only focuses on the mean estimation task, this should be addressed earlier, including the abstract, intro and even the title, as the paper does not study the general problem of federated learning.
2. Typo: In line 131, it should be " if agent $j$ has \emph{more} samples than ..."




**Time Spent Reviewing:**

4.5

---

> ### Author Response · Authors · 2021-08-09
> **Response to cBBM**
>
> Thank you for your comments, as well as your comment about defining terms - we will revise our paper to include more clear explanations. (Just to answer your question, we say that a player “weakly prefers” A to B” if that player either would rather have A, or if the player is ambivalent between A and B).
>
> This review also mentioned a question about how general our federated learning model is. It’s worth noting that our model, taken from the Donahue & Kleinberg paper, models both linear regression and mean estimation (when the number of points is much larger than the dimension of the linear regression problem). However, the broader point is a good one: we should be very clear that our paper is studying a model of federated learning that is simple enough to admit theoretical analysis. We will aim to revise the paper, especially the introduction and abstract, to clarify that this is a stylized model aiming to generate insights. Thank you for highlighting this point!

---

### Official Review · Reviewer_2WfY · 2021-07-17

**Rating:** 5
**Confidence:** 2

**Summary:**

This paper studies a basic federated learning problem, in which some agents, each one with their own dataset, have to jointly agree on a global model. The paper focuses on the trade-off between stability (enforcing that agents are incentivized to collaborate in the global model) and optimality (maximizing social welfare). The paper studies this trade-off by means of a Price of Anarchy analysis.

**Limitations And Societal Impact:**

Adequate.

**Main Review:**

Originality: The problem studied in the paper is interesting, being part of a rather new research filed that is receiving a growing attention from the literature in recent years. As far as I am concerned, the Price of anarchy analysis is novel in this field.

Quality: As far as I am concerned, all the proofs are correct.

Clarity: The paper has some major problem in the presentation. First, it is not clear from the first three sections which elements are introduced in this paper and which ones are taken from the literature. I think that re-organizing the sections so as to have one section entirely devoted to preliminaries and model definition would help improve clarity.

Significance: The approach of the paper to federated learning is mainly theoretical, I think that some more experimental analysis would be expected for a NeurIPS paper on this topic.

**Time Spent Reviewing:**

3

---

> ### Author Response · Authors · 2021-08-09
> **Response to 2WfY**
>
> Thank you for your comments! First, we agree that presenting contributions clearly is very important. In fact, Section 3.1 in our paper was intended to serve exactly the purpose you mention: within this section, we describe all of the results and models from prior work that we will use. All other results in the paper are original to our work. From this review, it seems that we could work to make this division clearer: we will aim to revise Section 3.1 to make it more clear that this is the purpose it is serving, and that the remaining results afterward are original to the current work.  We hope this will make the division in contributions even sharper.
>
> Regarding your question on experimental results, we first wish to note that none of our results, strictly speaking, need experiments. Because the Price of Anarchy is a worst-case bound, it already gives stronger results than any experiments could give. However, we agree  that experiments can provide insight in several ways: for example, computational experiments can give insight into the performance of the average-case instance, rather than the worst-case instance. In the process of writing this paper, we did have several computational experiments that helped to build intuition for the theoretical results. For example, as relates to reviewer WHyG’s question, we learned that every case we experimentally explored had an individually stable solution. We can plan to include these results, as well as code, alongside a revised version of this paper.

---

### Official Review · Reviewer_15pL · 2021-07-18

**Rating:** 6
**Confidence:** 4

**Summary:**

This paper examines the game theoretic aspect of federated learning. In the proposed model, the players are strategic and may not provide their data to the learning platform and choose to federate in a small coalition. The authors provide an algorithm to compute the optimal federating arrangement, and also analyze the bound of the price of anarchy in theory.

**Limitations And Societal Impact:**

Please refer to the main review for details.

**Main Review:**

This paper is written in a nice way and easy to understand. The concept of price of anarchy is worth investigating in this setting as a low price means the federtion is beneficial to the welfare. The theoretic results are interesting and obtained through careful analyses (did not check all the details though). However, I am concerned with one of the fundamental assumptions of the paper. Lemma 1, which is from another paper, assumes that the parameter \theta_j of player j is drawn from a common distribution. This paper also implicitly used the same assumption. Line 169 further assumes that n_i is fixed and known by all player. With this assumption, all players know the actual underlying data distribution and can just learn any model by simply sample enough points from this distribution. One of the implicit but fundamental assumptions of federated learning is that each player does not know the data distribution of others so they cannot get more useful data to train their own model. Can the authors provide more justifications to their modeling choice?

**Time Spent Reviewing:**

4.5

---

> ### Author Response · Authors · 2021-08-09
> **Response to 15pL**
>
> Thank you for your comments! Regarding your question on the assumptions, we think there may be a misunderstanding about the model. We agree that it is vital to get the assumptions of the model correct. This is why we started with standard high-level assumptions of federated learning: players cannot see data from other players and do not know the true generating distribution for data, but do know how many samples other players have, and may have estimates as to the level of noise in the sampling process and how dissimilar players might be from each other.
>
> To be slightly more specific, the model we use (from Donahue & Kleinberg) is a 2-stage model: first, players draw their true parameters from some (unknown) generating distribution. Then, each player draws their samples from a distribution governed by their true parameters - which are again unknown.  This is the sense in which players do not have access to the true distributions of their own data or the data of others; they only have indirect estimates of its properties.
>
> For example, consider the mean estimation case: suppose that each player draws a mean \theta_i from an exponential distribution with parameter lambda = 3 and similarly draws their noise level \epsilon_i^2 from some exponential distribution with parameter lambda =2. Wwith these parameters, mu_e = 1/2 and sigma^2 = 1/9. Then, having a true mean, each player draws their n_i data samples from a normal distribution N(\theta_i, \epsilon_i^2). Each player knows how many samples they and other players have: equivalent to knowing {n_i} for all i. They also know mu_e and sigma^2, but do *not* know the generating distributions (for example, they do not know that they are exponential). Additionally, they do not know their personal distributions (that they are normal distributions or which parameters they are using). As a result, it is impossible for any player to sample more data: they must rely on the samples they already have.
>
> We agree that it’s very important to make these assumptions clear, and these review comments are very helpful in showing how we can explain them more clearly.  Based on it, we will revise the paper to make sure our model and assumptions are more clearly stated.

---

> > ### Comment · Reviewer_15pL · 2021-08-30
> > **Thanks for the response**
> >
> > Thanks for the clarification. I understand that analyze the federated learning model from a game-theoretic perspective is difficult. Although some results still use those strong assumptions implicitly, I think this is an interesting model and I have raised my score to 6.

---

### Official Review · Reviewer_KY9P · 2021-07-18

**Rating:** 7
**Confidence:** 3

**Summary:**

This paper studies price of anarchy (PoA) in federated learning. Following the prior work of Donahue and Kleinberg [2021] which initiate a game theoretic model for federated learning, the authors propose a notion of optimality given the average error rates among agents. Under this notion of optimality, the authors first provide an efficient algorithm to compute the optimal solution, and then give PoA bounds that depend on parameters of settings.

**Limitations And Societal Impact:**

The reviewer is curious about whether the results can be generalized to other game-theoretic models of federated learning, e.g., for different notion of optimality and utility models of agents.

**Main Review:**

This paper is well-written, clear, and easy to follow. The problem studied is well-motivated from the context of federated learning and is important for practical applications. The collection of results seems novel and technically strong, and they give useful insight for real practice. The reviewer did not check all the proofs but they all look plausible in hindsight.

Comments:

1. In line 131-132, it seems something is missing in the sentence? Do you mean "player j has *more* or *less* samples than k"? The reviewer wonders whether the claim that "player j will have lower expected error than k" is always true in either situation. Doesn't it also depend on noise and dissimilarity of the distributions?

2. It seems that all the lemmas and theorems in section 4.1 highly depend on the modeling assumption, such as the notion of optimality and utility models of agents. How robust are these results? Could they be extended to more broader models?

3. The reviewer finds that the contrast of PoA bounds due to sample sizes v.s. noise and dissimilarity is very interesting. Could you provide more insights about these results? It seems that the results are driven by the fact that the only difference across agents is the number of samples they have (as their distributions are drawn from the same prior). It would be interesting to see what happen if their priors are different.

Minors:

1. Line 56: "leaves open" remove one of them
2. Line 237: Lemma 9 here?

**Time Spent Reviewing:**

2

---

> ### Author Response · Authors · 2021-08-09
> **Response to KY9P**
>
> Thank you for your comments! We will address them point-by-point below.
>
> Comment 1: Thank you for pointing this out: In the Donahue and Kleinberg model, it is always the case that a player with more samples will have lower error, but this is not always true in federated learning in general. We will revise this sentence to be more clear.
>
> Comment 2: Absolutely: the results do, in some cases, depend on the definition of optimality and the model of federated learning used. In Appendix A, we discuss an alternate notion of optimality. Given that it is a different definition of optimality, we would expect that the results would differ - from our preliminary analysis in this vein (which we didn’t include in this submission), it did seem that the results would differ somewhat. Specifically, because the appendix definition of optimality relies on an unweighted notion of cost, it results in an optimal arrangement that is often a better outcome for smaller players and worse outcome for larger players. Similarly, a different model of federated learning would potentially lead to different results in both stability and optimality - we will likely defer results about further categories of federated learning models for future work.
>
> Comment 3: This is also a very insightful question: the model we use does rely on the fact that players differ only in the number of samples they have, not any difference in true priors. We have explored an extension to this model where players are allowed to differ in the distribution their true parameters are drawn from. Specifically, each player has a “type” which relates to the true generating distribution that their parameters are drawn from. (As an example, consider that type A hospitals might be small and rural, while type B hospitals might be large and urban: hospitals of the same type might reasonably expect to be more similar to each other, even if they do not know the true generating distribution). Players know each other’s types and some rough parameters related to it, but do not (as before) know the full generating distribution. Because this formulation involves many more parameters (describing, for example, the average dissimilarity between groups as well as within groups), we opted to complete the PoA analysis with the simpler model in the paper.  However, we can plan to include the summary of the more complex model in an appendix of a revised version of the paper.

---

### Official Review · Reviewer_WHyG · 2021-07-25

**Rating:** 7
**Confidence:** 3

**Summary:**

The paper studies a coalition formation process in federated learning. When agents form a coalition, they aggregate their learned parameters in the hope of a more accurate result. The paper presented an algorithm to obtain the optimal way of partitioning the agents in coalitions, so that the total weighted error of the agents is minimized. In addtion, the paper also analyzed the price of anarchy and provided upper bounds of it.

**Limitations And Societal Impact:**

Adequately addressed.

**Main Review:**

The paper presented some interesting and novel results. The topic about incentives in federated learning also looks important for this research area and is likely to inspire new ideas.

The paper is well-written. The presentation is mostly clear and the results look sound (though I didn't check details in the appendix).

One question I have is about the emptiness of the core: Does a core (or individual) stable arrangement always exist? If this is not guaranteed, I wonder how meaningful the PoA bounds are.

Several typos:

- 131, Page 3, "...if player j has samples than..." --> if player j has more samples?

- 205, Page 5, "could could"

- 216, Page 5, there's a grammar issue.

**Time Spent Reviewing:**

4

---

> ### Author Response · Authors · 2021-08-09
> **Response to WHyG**
>
> Thank you for your comments and for catching so many typos!
>
> One question you asked was about whether a core or individually stable solution always exists. In general, a stable solution is not guaranteed to exist (the Donahue & Kleinberg paper studies stability in more detail). However, it’s worth noting two points. Firstly, the PoA bound is still useful even if a stable solution doesn’t exist. Stable solutions with high cost are actually undesirable, in our formulation, because they may draw players away from the optimal (but unstable) arrangement. In the case that no stable solution exists, this danger isn’t present, so the PoA bound isn’t needed.
>
> Secondly, this question relates in a sense to our answer to 2WfY’s question on experimental results. In the process of writing this paper, we wrote code and ran many simulations related to this model. In the course of doing so, every single example we found had at least one individually stable solution.  As a result, even though stable solutions are not guaranteed to exist, our computational experiments show that stable solutions appear to be nearly ubiquitous in practice, and hence a relevant construct for beginning PoA analysis.  In a revised version, we would include this code, which would potentially help clarify the extent to which stability can be expected.

---

> > ### Comment · Reviewer_WHyG · 2021-08-31
> > **Thank you for your reply**
> >
> > Thanks. When there is no stable solution, I think the danger still exists. E.g., the system could oscillate between states with high costs. I think your second point provides a stronger argument, and I'm convinced overall.

---

### Author Response · Authors · 2021-08-09
**General response**

We wish to thank all of the reviews for their thoughtful, kind, and helpful comments on our paper, especially during an ongoing pandemic.

---

### Decision · Program_Chairs · 2021-09-27

**Decision:**

Accept (Poster)

**Comment:**

The rebuttals helped the Reviewers to understand better the original contributions provided in the paper. However, I have two major suggestions for the authors when producing the camera-ready version of the paper:
1. clarify better the assumptions, even those made implicitly, and clarifying them in the formal statements;
2. re-organize the first three sections to clarify which are the original contributions and which are related works, in particular it would be nice to have one section entirely devoted to preliminaries and model definition.